# A hierarchy of cell death pathways confers layered resistance to shigellosis in mice

Justin L Roncaioli[1], Janet Peace Babirye[1], Roberto A Chavez[1], Fitty L Liu[1], Elizabeth A Turcotte[1], Angus Y Lee[2], Cammie F Lesser[3,4,5], Russell E Vance[1,2,6,7]*

[1]Division of Immunology & Molecular Medicine, Department of Molecular & Cell Biology, University of California, Berkeley, Berkeley, United States; [2]Cancer Research Laboratory, University of California, Berkeley, Berkeley, United States; [3]Department of Microbiology, Harvard Medical School, Boston, United States; [4]Broad Institute of Harvard and MIT, Cambridge, United States; [5]Department of Medicine, Division of Infectious Diseases, Massachusetts General Hospital, Boston, United States; [6]Immunotherapeutics and Vaccine Research Initiative, University of California, Berkeley, Berkeley, United States; [7]Howard Hughes Medical Institute, University of California, Berkeley, Berkeley, United States

*For correspondence:
rvance@berkeley.edu

**Abstract** Bacteria of the genus *Shigella* cause shigellosis, a severe gastrointestinal disease driven by bacterial colonization of colonic intestinal epithelial cells. Vertebrates have evolved programmed cell death pathways that sense invasive enteric pathogens and eliminate their intracellular niche. Previously we reported that genetic removal of one such pathway, the NAIP–NLRC4 inflammasome, is sufficient to convert mice from resistant to susceptible to oral *Shigella flexneri* challenge (Mitchell et al., 2020). Here, we investigate the protective role of additional cell death pathways during oral mouse *Shigella* infection. We find that the Caspase-11 inflammasome, which senses *Shigella* LPS, restricts *Shigella* colonization of the intestinal epithelium in the absence of NAIP–NLRC4. However, this protection is limited when *Shigella* expresses OspC3, an effector that antagonizes Caspase-11 activity. TNFα, a cytokine that activates Caspase-8-dependent apoptosis, also provides potent protection from *Shigella* colonization of the intestinal epithelium when mice lack both NAIP–NLRC4 and Caspase-11. The combined genetic removal of Caspases-1, -11, and -8 renders mice hypersusceptible to oral *Shigella* infection. Our findings uncover a layered hierarchy of cell death pathways that limit the ability of an invasive gastrointestinal pathogen to cause disease.

## Editor's evaluation

This paper provides important new information on the role of cellular death pathways in mediating resistance and susceptibility of mice to experimental shigellosis. The results rely on experimental observations on the outcome of Shigella in mice gene deficiencies and are convincing. The results will be of interest to immunologists, cell biologists and infectious disease researchers.

## Introduction

*Shigella* is a genus of enteric bacterial pathogens that causes ~270 million yearly cases of shigellosis, with ~200,000 of these resulting in death (*Khalil et al., 2018*). Shigellosis manifests as an acute inflammatory colitis resulting in abdominal cramping, fever, and in severe cases, bloody diarrhea (dysentery) (*Kotloff et al., 2018*). Bacterial invasion of the colonic intestinal epithelium and subsequent

dissemination between adjacent intestinal epithelial cells (IECs) is believed to drive inflammation and disease. *Shigella* pathogenesis is mediated by a virulence plasmid which encodes a type three secretion system (T3SS) and more than 30 virulence factors or effectors (*Schnupf and Sansonetti, 2019*; *Schroeder and Hilbi, 2008*). The T3SS injects effectors into the host cell to facilitate bacterial invasion, escape into the cytosol, and disarmament of the host innate immune response to make the cytosol a hospitable niche for replicating *Shigella* (*Ashida et al., 2015*). The virulence plasmid also encodes IcsA, a bacterial surface protein that facilitates cytosolic actin-based motility and is essential for bacterial spread to neighboring IECs (*Bernardini et al., 1989*; *Goldberg and Theriot, 1995*; *Mattock and Blocker, 2017*).

The innate immune system can counteract intracellular bacterial pathogens by inducing programmed cell death (*Williams, 1994*). Programmed cell death eliminates the intracellular pathogen niche, maintains epithelial barrier integrity, promotes clearance of damaged cells, and enhances presentation of foreign antigens to cells of the adaptive immune system (*Deets et al., 2021*; *Doran et al., 2020*; *Jorgensen et al., 2017*; *Koch and Nusrat, 2012*; *Yatim et al., 2017*). Three main modes of programmed cell death are common to mammalian cells: pyroptosis, apoptosis, and necroptosis. Each is controlled by distinct sensors and conserved downstream executors which together provide a formidable barrier that pathogens must avoid or subvert for successful intracellular replication. Of particular relevance to *Shigella* and other gastrointestinal pathogens, cell death of IECs is accompanied by a unique cellular expulsion process that rapidly and selectively ejects dying or infected cells from the epithelial layer, thereby potently limiting pathogen invasion into deeper tissue (*Fattinger et al., 2021*; *Knodler et al., 2014*; *Rauch et al., 2017*; *Sellin et al., 2014*).

*Shigella* is an example of a pathogen in intense conflict with host cell death pathways (*Ashida et al., 2021*). *Shigella* encodes multiple effectors to prevent cell death in human cells, including OspC3 to block Caspase-4 inflammasome activation (*Kobayashi et al., 2013*; *Li et al., 2021*; *Mou et al., 2018*; *Oh et al., 2021*), IpaH7.8 to inhibit Gasdermin D-dependent pyroptosis (*Luchetti et al., 2021*), OspC1 to suppress Caspase-8-dependent apoptosis (*Ashida et al., 2020*), and OspD3 to block necroptosis (*Ashida et al., 2020*). The antagonism of these pathways (and perhaps others that are yet undiscovered) and the resulting maintenance of the epithelial niche appears sufficient to render humans susceptible to *Shigella* infection. Mice, however, are resistant to oral *Shigella* challenge because *Shigella* is unable to counteract epithelial NAIP–NLRC4-dependent cell death and expulsion (*Chang et al., 2013*; *Mitchell et al., 2020*). Removal of the NAIP–NLRC4 inflammasome renders mice susceptible to shigellosis, providing a tractable genetic model to dissect *Shigella* pathogenesis after oral infection in vivo (*Mitchell et al., 2020*).

Here, we use the NAIP–NLRC4-deficient mouse model of shigellosis to investigate the role of programmed cell death in defense against *Shigella* in vivo. We find that Caspase-11 (CASP11), a cytosolic sensor of LPS and the mouse ortholog of human Caspase-4 (*Shi et al., 2014*), provides modest protection from *Shigella* infection in the absence of NAIP–NLRC4. As in humans, this pathway is antagonized by the *Shigella* effector OspC3, and genetic removal of *ospC3* from *Shigella* results in a significant CASP11-dependent reduction in bacterial colonization of IECs and virulence. We also find that TNFα, a cytokine that can induce TNF receptor 1 (TNFRI)-dependent extrinsic apoptosis (*Piguet et al., 1998*), defends mouse IECs from bacterial colonization and limits subsequent disease. TNFα-dependent protection is strongest when mice lack both NAIP–NLRC4 and CASP11, revealing a hierarchical program of cell death pathways that counteract *Shigella* in vivo. *Casp1/11⁻/⁻Ripk3⁻/⁻* and *Casp8⁻/⁻Ripk3⁻/⁻* mice, which lack some but not all key components of pyroptosis, apoptosis, and necroptosis, are largely protected from disease, revealing redundancies among these pathways. *Casp1/11/8⁻/⁻Ripk3⁻/⁻* mice, however, are hyper-susceptible to shigellosis, indicating that programmed cell death is a predominant host defense mechanism against *Shigella* infection. Furthermore, neither interleukin-1 receptor (IL-1R)-mediated signaling nor myeloid-restricted NAIP–NLRC4 have an apparent effect on *Shigella* pathogenesis, suggesting that it is cell death of IECs that primarily protects mice from shigellosis. Our findings underscore the importance of cell death in defense against intracellular bacterial pathogens and provide an example of how layered and hierarchical immune pathways can provide robust defense against pathogens that have evolved a broad arsenal of virulence factors.

# Results

## CASP11 contributes to resistance of B6 versus 129 *Nlrc4*$^{-/-}$ mice to shigellosis

We previously generated NLRC4-deficient mice on the 129S1/SvImJ (129) background (129.*Nlrc4*$^{-/-}$) and observed that these mice appeared more susceptible to oral *Shigella flexneri* challenge than C57BL/6J (B6) NLRC4-deficient mice (B6.*Nlrc4*$^{-/-}$) (*Mitchell et al., 2020*). We reasoned that the apparent difference between the strains might be due to genetic and/or microbiota differences. To address these possibilities, we infected co-housed B6.*Nlrc4*$^{-/-}$ and 129.*Nlrc4*$^{-/-}$ mice and directly compared disease severity between the two strains (*Figure 1*, light blue versus pink symbols). The B6.*Nlrc4*$^{-/-}$ mice exhibited only modest weight loss (5–10% of starting weight) through two days and began to recover by day 3 (*Figure 1A*). The 129.*Nlrc4*$^{-/-}$ mice, however, continued to lose weight through day 3 (10–15% of starting weight) (*Figure 1A*). Upon sacrifice at day 3, we harvested the IEC fraction from the cecum and colon of each mouse, washed this fraction in gentamicin to eliminate any extracellular *Shigella*, and lysed these cells to enumerate intracellular bacterial colonization of IECs. IECs from 129.*Nlrc4*$^{-/-}$ mice harbored >10-fold higher intracellular *Shigella* burdens than those from B6.*Nlrc4*$^{-/-}$ mice (*Figure 1B*). We also found that 129.*Nlrc4*$^{-/-}$ mice had higher levels of inflammatory cytokines CXCL1 and IL-1β in their intestinal tissue, as measured by ELISA (*Figure 1C and D*). CXCL1 and IL-1β are NF-κB-induced cytokines previously implicated in driving disease during shigellosis by initiating inflammation and promoting innate immune cell recruitment to the gut (*Arondel et al., 1999*; *Sansonetti et al., 1999*; *Sansonetti et al., 2000*; *Singer and Sansonetti, 2004*). Here, these cytokines serve as biomarkers of disease. Because the ELISA used cannot distinguish between pro-IL-1β and cleaved IL-1β, reported IL-1β levels reflect the strength of the NF-κB response (as do reported CXCL1 levels) rather than the strength of Caspase-1 activation. The 129.*Nlrc4*$^{-/-}$ mice also exhibited significantly more gross cecum shrinkage than B6.*Nlrc4*$^{-/-}$ mice (*Figure 1E*) and there were modest but insignificant increases in diarrhea (as measured by the wet weight to dry weight ratio of mouse feces) in 129.*Nlrc4*$^{-/-}$ mice relative to the B6.*Nlrc4*$^{-/-}$ mice at 2 and 3 days post-infection (*Figure 1F*). We scored mouse feces for the presence of occult blood (score = 1) or macroscopic blood (score = 2) at days 2 and 3, the sum of which represents a blood score from 0 to 4 (*Figure 1G*). All 129.*Nlrc4*$^{-/-}$ mice had occult blood in their feces on at least one of these days, with many having occult or macroscopic blood on both days. In contrast, B6.*Nlrc4*$^{-/-}$ mice did not exhibit fecal blood.

The significant difference in disease severity between co-housed 129 and B6 *Nlrc4*$^{-/-}$ mice suggested that genetic rather than microbiota differences might explain the differential susceptibility of the strains. The mouse non-canonical inflammasome Caspase-11 and its human orthologs Caspases-4 and -5 sense cytosolic *Shigella* LPS to initiate pyroptosis (*Hagar et al., 2013*; *Kayagaki et al., 2011*; *Kobayashi et al., 2013*; *Shi et al., 2014*). Notably, 129 mice are naturally deficient for Caspase-11 (*Kayagaki et al., 2011*). To determine if Caspase-11 contributes to the difference in susceptibility between these strains, we crossed B6.*Nlrc4*$^{-/-}$ and 129.*Nlrc4*$^{-/-}$ mice to generate B6/129.*Nlrc4*$^{-/-}$ F$_1$ hybrids (*Figure 1—figure supplement 1A*). We infected these F$_1$ B6/129. *Nlrc4*$^{-/-}$ hybrids and found that they were relatively resistant to *Shigella* challenge and their disease profile more consistently resembled that of the parental B6.*Nlrc4*$^{-/-}$ mice (*Figure 1—figure supplement 1B–F*). These results are consistent with the possibility that a dominant gene on the C57BL/6J background provides protection from *Shigella*. Next, we backcrossed these hybrids to the 129.*Nlrc4*$^{-/-}$ parental strain to generate littermate *Nlrc4*$^{-/-}$ mice that were homozygous 129/129 or heterozygous B6/129 at *Casp11* (*Figure 1—figure supplement 1A*). These *Nlrc4*$^{-/-}$ backcrossed mice were co-housed with their parental 129.*Nlrc4*$^{-/-}$ and B6.*Nlrc4*$^{-/-}$ strains for >3 weeks, infected with *Shigella*, and genotyped at the *Casp11* locus to determine whether a functional B6 *Casp11* allele would correlate with reduced disease severity.

Indeed, backcrossed *Nlrc4*$^{-/-}$ mice that were heterozygous B6/129 at *Casp11* (*Figure 1*, maroon symbols) were more resistant to shigellosis than backcrossed *Nlrc4*$^{-/-}$ mice that were 129/129 at *Casp11* (*Figure 1*, dark blue symbols). Mice that were heterozygous B6/129 at *Casp11* showed a similar weight loss pattern to the parental B6.*Nlrc4*$^{-/-}$ mice and began to recover by day 3 while the weight loss in mice that were homozygous 129/129 at *Casp11* phenocopied that of the parental 129.*Nlrc4*$^{-/-}$ mice (*Figure 1A*). Consistent with these results, mice that were homozygous 129/129 at *Casp11* also exhibited significantly enhanced bacterial colonization of the intestinal epithelium (*Figure 1B*). We observed trending but insignificant increases in inflammatory cytokine CXCL1 (*Figure 1C*) and cecum

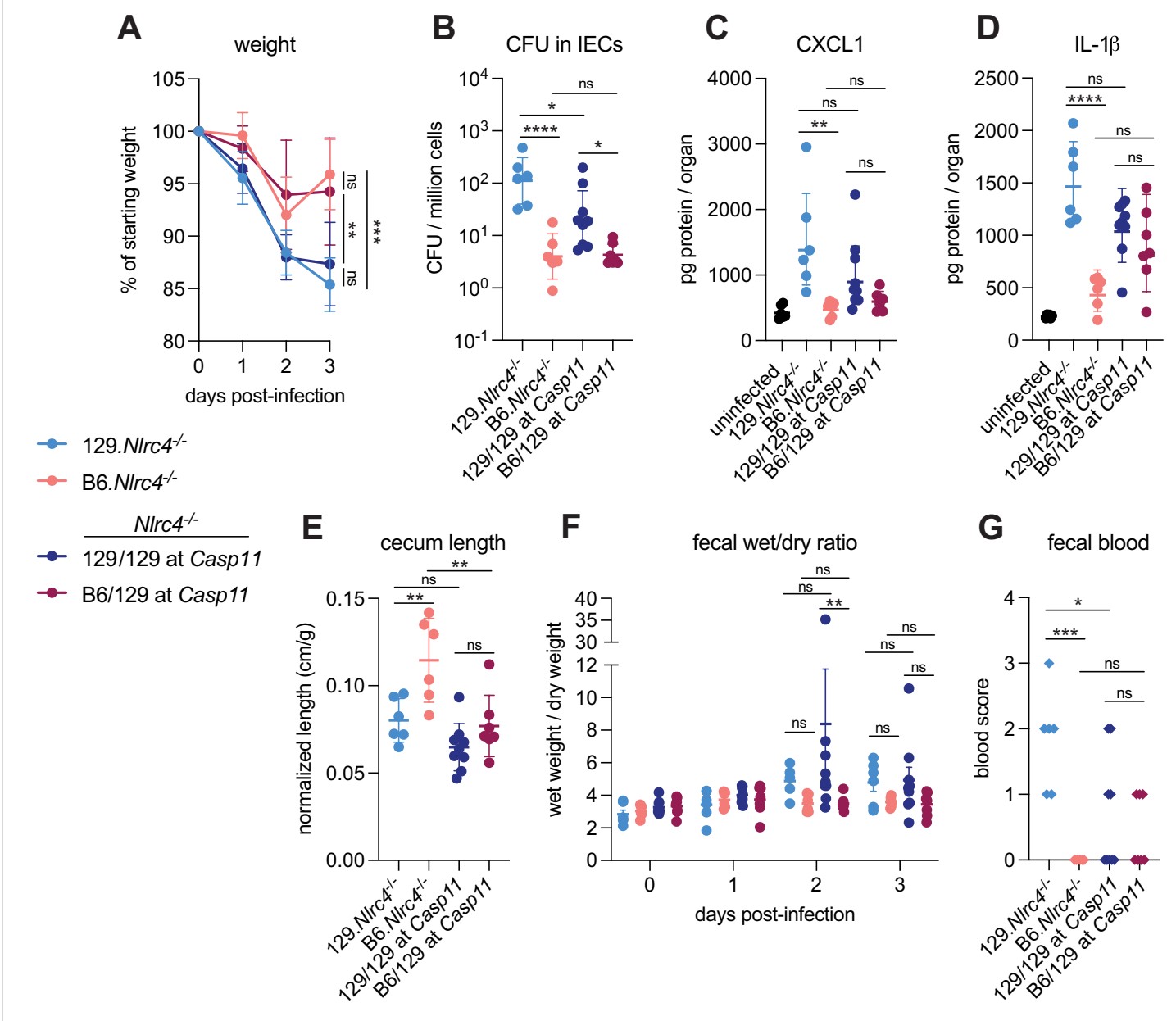

**Figure 1.** CASP11 contributes to resistance of B6 versus 129 *Nlrc4⁻/⁻* mice to shigellosis. (A–G) B6.*Nlrc4⁻/⁻* mice (pink, n=6), 129.*Nlrc4⁻/⁻* mice (light blue, n=6), and backcrossed littermates that are homozygous 129/129 at *Casp11* (dark blue, n=9) or heterozygous B6/129 at *Casp11* (maroon, n=7) were co-housed for 3 weeks, treated orally with 25 mg streptomycin sulfate in water, and orally challenged the next day with 10⁷ colony forming units (CFUs) of wild-type (WT) *Shigella flexneri*. Mice were sacrificed at 3 days post-infection. (A) Mouse weights from 0 through 3 days post-infection. Each symbol represents the mean for all mice of the indicated genotype. (B) *Shigella* CFUs per million cells from the combined intestinal epithelial cell (IEC) enriched fraction of gentamicin-treated cecum and colon tissue. (C, D) CXCL1 and IL-1β levels measured by ELISA from homogenized cecum and colon tissue of infected mice. (E) Quantification of cecum lengths normalized to mouse weight prior to infection; cecum length (cm)/mouse weight (g). (F) The ratio of fecal pellet weight when wet (fresh) divided by the fecal pellet weight after overnight drying. A larger wet/dry ratio indicates increased diarrhea. Pellets were collected daily from 0 to 3 days post-infection. (G) Additive blood scores from feces collected at 2 and 3 days post-infection. 1=occult blood, 2=macroscopic blood for a given day, maximum score is 4. (B–G) Each symbol represents one mouse. Data collected from one experiment. Mean ± SD is shown in (A, C– E). Geometric mean ± SD is shown in (B). Mean ± SEM is shown in (F). Statistical significance was calculated by one-way ANOVA with Tukey's multiple comparison test (A (day 3), B, C, D, E, and G) and by two-way ANOVA with Tukey's multiple comparison test (F). Data were log-transformed prior to calculations in (B) and (F) to achieve normality. *p<0.05, **p<0.01, ***p<0.001, ****p<0.0001, ns = not significant (p>0.05).

The online version of this article includes the following figure supplement(s) for figure 1:

**Figure supplement 1.** B6/129.*Nlrc4⁻/⁻* F₁ hybrids are modestly susceptible to *Shigella*.

**Figure supplement 2.** *Hiccs* does not contribute to resistance of B6 versus 129 *Nlrc4⁻/⁻* mice to shigellosis.

shrinkage (*Figure 1E*) and significantly more pronounced diarrhea at day 2 (*Figure 1F*) in mice that were 129/129 at *Casp11* relative to mice that were B6/129 at *Casp11*. Despite these differences, there was no strong correlation between IL-1β levels (*Figure 1D*) or fecal blood score (*Figure 1G*) and *Casp11* genotype, suggesting that while *Casp11* contributes to resistance, there are additional genetic modifiers present on the 129 or B6 background that affect susceptibility to shigellosis. As these additional modifiers appear to be relatively weak compared to *Casp11*, we did not attempt to map them genetically. However, we did specifically test for a contribution of *Hiccs*, a genetic locus in 129 mice that associates with increased susceptibility to *Helicobacter hepaticus*-dependent colitis (*Boulard et al., 2012*). To do so, we genotyped the same *Nlrc4⁻/⁻* backcrossed mice at the *Hiccs* locus and used the same data from *Figure 1* to determine whether a 129 or B6 *Hiccs* allele associated with differences in disease (*Figure 1—figure supplement 1A*). In contrast to *Casp11*, we found that *Hiccs* did not significantly correlate with increased susceptibility to shigellosis (*Figure 1—figure supplement 2*).

## CASP11 modestly contributes to resistance of B6.*Nlrc4⁻/⁻* mice to shigellosis

To define the role of mouse Caspase-11 in a uniform genetic background, we generated *Casp11⁻/⁻* mice on the B6.*Nlrc4⁻/⁻* background using CRISPR-Cas9 editing (*Figure 2—figure supplement 1*). We previously found that *Casp1/11⁻/⁻* mice are resistant to oral wild-type (WT) *S. flexneri* infection, likely because NLRC4-dependent Caspase-8 activation is sufficient to prevent bacterial colonization of IECs (*Figure 2—figure supplement 2*; *Mitchell et al., 2020*; *Rauch et al., 2017*). Thus, Caspase-11 is dispensable for protection from WT *Shigella* challenge when mice express functional NLRC4, but Caspase-11 could still be critical as a backup pathway in the absence of NLRC4. We therefore challenged B6.*Nlrc4⁻/⁻Casp11⁺/⁻* and B6.*Nlrc4⁻/⁻Casp11⁻/⁻* littermates with WT *Shigella* and assessed pathogenicity for 2 days following infection.

We observed a modest increase in susceptibility to *Shigella* infection in B6.*Nlrc4⁻/⁻Casp11⁻/⁻* mice relative to B6.*Nlrc4⁻/⁻Casp11⁺/⁻* (*Figure 2*). While B6.*Nlrc4⁻/⁻Casp11⁻/⁻* mice did not experience more weight loss (*Figure 2A*), cecum shrinkage (*Figure 2B*), or diarrhea (*Figure 2C*) than B6.*Nlrc4⁻/⁻Casp11⁺/⁻*, there was a fivefold increase in *Shigella* burdens in IECs from B6.*Nlrc4⁻/⁻Casp11⁻/⁻* mice (*Figure 2D*), indicating that Caspase-11 protects the mouse epithelium from bacterial colonization in the absence of NLRC4. Intestinal tissue from B6.*Nlrc4⁻/⁻Casp11⁻/⁻* mice also expressed significantly higher levels of CXCL1 than tissue from B6.*Nlrc4⁻/⁻Casp11⁺/⁻* (*Figure 2E*). IL-1β levels appeared elevated in B6.*Nlrc4⁻/⁻Casp11⁻/⁻* mice relative to B6.*Nlrc4⁻/⁻Casp11⁺/⁻*, however this difference was not significant (*Figure 2F*). B6.*Nlrc4⁻/⁻Casp11⁺/⁻* did not exhibit blood in their feces but two of the nine B6.*Nlrc4⁻/⁻Casp11⁻/⁻* did present with occult blood (*Figure 2G*) – an increase that is not statistically significant. These results suggest that Caspase-11 has a relatively modest contribution to defense against WT *Shigella*. Indeed, a minor role for Caspase-11 is expected given that *Shigella* is known to encode an effector called OspC3 that inhibits Caspase-11 (see below). Nevertheless, taken together, our results in mixed 129/B6.*Nlrc4⁻/⁻* and B6.*Nlrc4⁻/⁻Casp11⁻/⁻* mice indicate that Caspase-11 contributes to defense against *Shigella* in vivo as a backup pathway in the absence of NLRC4 (*Figure 2—figure supplement 2*).

## *Shigella* effector OspC3 is critical for virulence in oral *Shigella* infection

*S. flexneri* protein OspC3 is a T3SS-secreted effector that inhibits both human Caspase-4 and mouse Caspase-11 to suppress pyroptosis (*Kobayashi et al., 2013*; *Li et al., 2021*; *Mou et al., 2018*; *Oh et al., 2021*). While OspC3 has been shown to be required for virulence during intraperitoneal mouse infection by *S. flexneri* (*Li et al., 2021*; *Oh et al., 2021*) and for intestinal colonization by *S. sonnei* in WT mice (*Alphonse et al., 2022*), the role of this effector has not been studied in an oral mouse model of infection where *Shigella* invades and replicates within the intestinal epithelium. Indeed, our results indicating a role for Caspase-11 in defense against WT *Shigella* (see above, *Figures 1 and 2*) suggested that the inhibition of Caspase-11 by OspC3 could be incomplete in epithelial cells. To test the role of OspC3 in shigellosis, we orally infected B6.*Nlrc4⁻/⁻* mice (a mixture of Caspase-11 sufficient co-housed B6.*Nlrc4⁻/⁻Casp11⁺/⁺* mice and B6.*Nlrc4⁻/⁻Casp11⁺/⁻* mice) with WT *S. flexneri* or a mutant stain that lacks OspC3 (Δ*ospC3*) (*Figure 3*). Consistent with our previous experiments, B6.*Nlrc4⁻/⁻* mice challenged with WT S*higella* developed shigellosis characterized by significant weight loss, increases in bacterial colonization of the intestinal epithelium, cecum shrinkage, diarrhea, and inflammatory

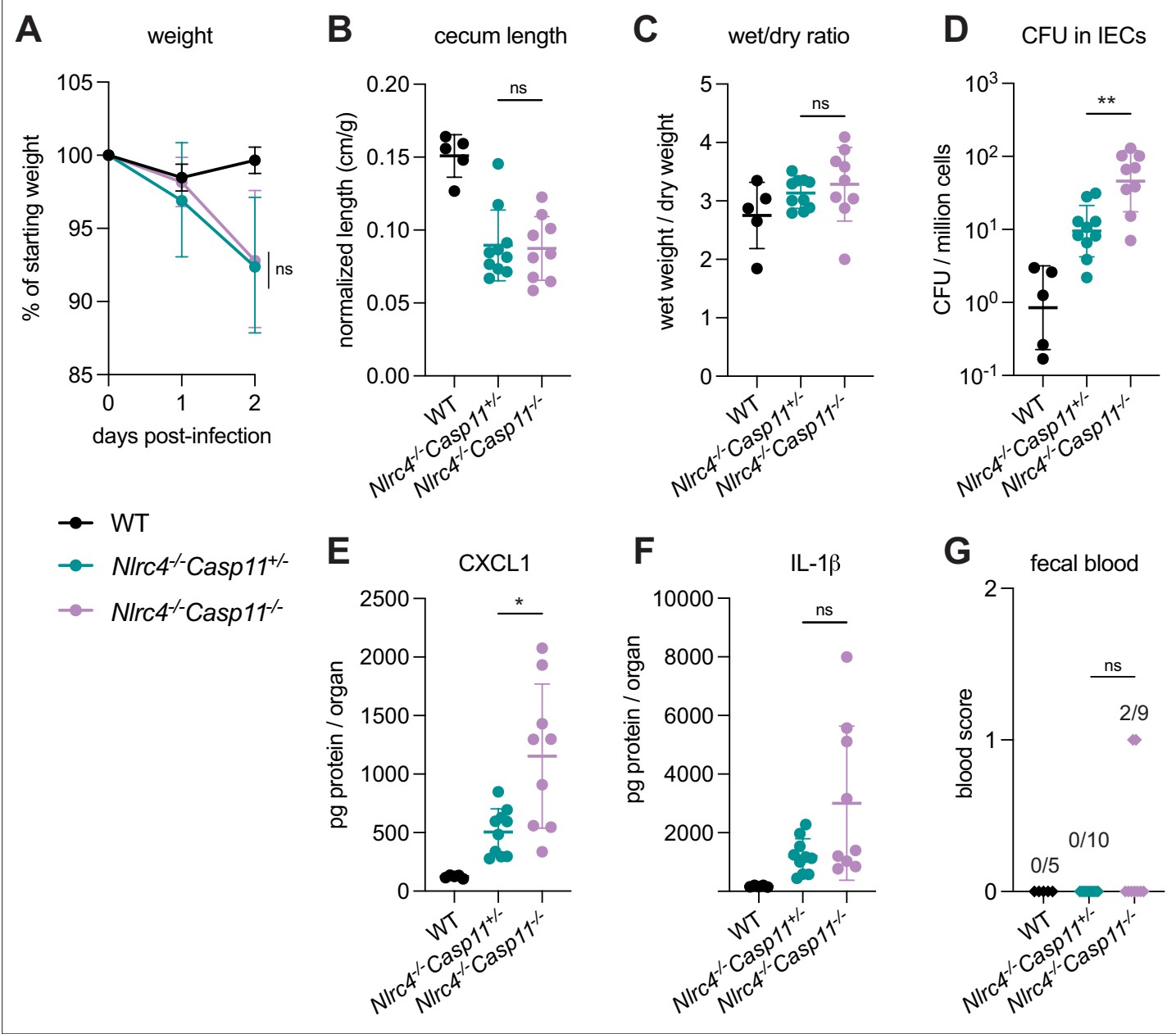

**Figure 2.** CASP11 modestly contributes to resistance of B6.*Nlrc4*⁻/⁻ mice to shigellosis. (**A–G**) B6.WT mice (co-housed B6.WT and B6.*Nlrc4*⁺/⁻*Casp11*⁺/⁻ mice, black, n=5) and B6.*Nlrc4*⁻/⁻*Casp11*⁺/⁻ (teal, n=10) and B6.*Nlrc4*⁻/⁻*Casp11*⁻/⁻ (lavender, n=9) littermates were treated orally with 25 mg streptomycin sulfate in water and orally challenged the next day with 10⁷ colony forming units (CFUs) of wild-type (WT) *Shigella flexneri*. Mice were sacrificed at 2 days post-infection. (**A**) Mouse weights from 0 through 2 days post-infection. Each symbol represents the mean for all mice of the indicated genotype. (**B**) Quantification of cecum lengths normalized to mouse weight prior to infection; cecum length (cm)/mouse weight (g). (**C**) The ratio of fecal pellet weight when wet (fresh) divided by the fecal pellet weight after overnight drying. Pellets were collected at day 2 post-infection. A larger wet/dry ratio indicates increased diarrhea. (**D**) *Shigella* CFUs per million cells from the combined intestinal epithelial cell (IEC) enriched fraction of gentamicin-treated cecum and colon tissue. (**E, F**) CXCL1 and IL-1β levels measured by ELISA from homogenized cecum and colon tissue of infected mice. (**G**) Blood scores from feces collected at 2 days post-infection. 1=occult blood, 2=macroscopic blood. (**B–G**) Each symbol represents one mouse. Data collected from two independent experiments. Mean ± SD is shown in (**A–C, E, F**). Geometric mean ± SD is shown in (**D**). Statistical significance was calculated by Mann-Whitney test in (**A–F**) and by Fisher's exact test in (**G**) where data were stratified by presence (score = 1 or 2) or absence (score = 0) of blood. In (**A**) statistical analysis was performed at day 2. *p<0.05, **p<0.01, ***p<0.001, ****p<0.0001, ns = not significant (p>0.05).

The online version of this article includes the following source data and figure supplement(s) for figure 2:

**Figure supplement 1.** B6.*Nlrc4*⁻/⁻*Casp11*⁻/⁻ mice have a 13 bp deletion in *Casp11* and loss of CASP11 protein.

**Figure supplement 1—source data 1.** Raw images of CASP11 protein western blots from bone marrow-derived macrophage lysates from mice edited

*Figure 2 continued on next page*

*Figure 2 continued*

at *Casp11* that are either heterozygous (left two lanes) or homozygous knockout (right two lanes).

**Figure supplement 2.** *Shigella* activates mouse cell death pathways.

cytokines (**Figure 3A–G**) relative to WT mice infected with WT *Shigella*. However, B6.*Nlrc4*⁻/⁻ mice challenged with Δ*ospC3 S. flexneri* were less susceptible to infection (**Figure 3**), exhibiting significantly less weight loss (**Figure 3B**), a >10-fold decrease in IEC colonization (**Figure 3C**), reduced cecum shrinkage (**Figure 3D**), and a decrease in CXCL1 (**Figure 3F**) relative to WT-infected B6.*Nlrc4*⁻/⁻ mice. We did not observe significant differences in diarrhea (**Figure 3E**) and IL-1β (**Figure 3G**) between these two groups. Interestingly, Δ*ospC3*-infected B6.*Nlrc4*⁻/⁻ mice did experience trending but insignificant increases in weight loss (**Figure 3B**), bacterial colonization of IECs (**Figure 3C**), cecum shrinkage (**Figure 3A and D**), and inflammatory cytokines (**Figure 3F and G**) relative to WT mice infected with WT *Shigella*. B6.*Nlrc4*⁻/⁻ mice infected with Δ*ospC3 S. flexneri* did not display fecal blood while six of the eleven B6.*Nlrc4*⁻/⁻ mice infected with WT *Shigella* did present with fecal blood (**Figure 3H**). These results indicate that Δ*ospC3 Shigella* is significantly attenuated in our B6.*Nlrc4*⁻/⁻ mouse model of shigellosis.

OspC3 directly inactivates mouse Caspase-11 (**Li et al., 2021**) but has also been reported to modulate other signaling pathways, including interferon signaling (**Alphonse et al., 2022**). To test if the effect of OspC3 on virulence is dependent on inhibition of mouse Caspase-11, we infected both B6.*Nlrc4*⁻/⁻ mice (that were a mixture of co-housed B6.*Nlrc4*⁻/⁻*Casp11*⁺/⁻ and B6.*Nlrc4*⁻/⁻*Casp11*⁺/⁺ mice) and B6.*Nlrc4*⁻/⁻*Casp11*⁻/⁻ mice (littermates with B6.*Nlrc4*⁻/⁻*Casp11*⁺/⁻) with either WT or Δ*ospC3 Shigella* strains. We again observed that *the ospC3* mutant was attenuated relative to WT *Shigella* in B6.*Nlrc4*⁻/⁻ mice (**Figure 4**). However, both WT and Δ*ospC3 Shigella* caused severe disease in B6.*Nlrc4*⁻/⁻*Casp11*⁻/⁻ mice, with comparable weight loss, bacterial colonization of the intestinal epithelium, cecum lengths, diarrhea, and fecal blood (**Figure 4A–D and G**). Δ*ospC3*-infected B6.*Nlrc4*⁻/⁻ mice exhibited significantly less weight loss, bacterial burdens, cecum shrinkage, and IL-1β relative to Δ*ospC3*-infected B6.*Nlrc4*⁻/⁻*Casp11*⁻/⁻ mice (**Figure 4A, B, C and F**). These results indicate that Caspase-11 is the primary physiological target of OspC3 in vivo. Caspase-11 provides potent defense against *Shigella* in the absence of OspC3, although this does not appear sufficient to fully compensate for the loss of NLRC4, as Δ*ospC3*-infected B6.*Nlrc4*⁻/⁻ mice exhibit a phenotype that trends toward modest susceptibility relative to WT-infected WT control mice (**Figure 3**, **Figure 4**). We did observe a trending but insignificant decrease in CXCL1 and a significant decrease in IL-1β in Δ*ospC3*-infected B6.*Nlrc4*⁻/⁻*Casp11*⁻/⁻ mice relative to WT-infected B6.*Nlrc4*⁻/⁻*Casp11*⁻/⁻ mice (**Figure 4E and F**), indicating that OspC3 might also affect immune pathways independent of Caspase-11. Again, we only observed modest differences in disease hallmarks between B6.*Nlrc4*⁻/⁻ and B6.*Nlrc4*⁻/⁻*Casp11*⁻/⁻ mice infected with WT *Shigella* (**Figure 4**), none of which were significant, consistent with the ability of OspC3 to significantly reduce Caspase-11 activity. These results confirm prior reports that OspC3 inhibits Caspase-11 in vivo (**Kobayashi et al., 2013**; **Li et al., 2021**; **Mou et al., 2018**; **Oh et al., 2021**) and further show that OspC3-dependent inhibition of Caspase-11 is required for *Shigella* virulence. Nonetheless, this inhibition is likely incomplete, as Caspase-11 still provides a small degree of protection in B6.*Nlrc4*⁻/⁻ mice even when *Shigella* expresses OspC3 (**Figures 2 and 4**).

## Neither myeloid NLRC4 nor IL-1 affects *Shigella* pathogenesis

The generally accepted model of *Shigella* pathogenesis proposes that *Shigella* bacteria cross the colonic epithelium via transcytosis through M-cells (**Schnupf and Sansonetti, 2019**; **Schroeder and Hilbi, 2008**). After transcytosis, *Shigella* is then believed to be phagocytosed by macrophages, followed by two additional steps: (1) the inflammasome-dependent lysis of infected macrophages to release bacteria to facilitate epithelial invasion (**Schnupf and Sansonetti, 2019**; **Suzuki et al., 2007**; **Zychlinsky et al., 1994**; **Zychlinsky et al., 1996**), and (2) the concomitant processing and release of IL-1β, a pro-inflammatory cytokine, that drives inflammation (**Arondel et al., 1999**; **Sansonetti et al., 1995**; **Sansonetti et al., 2000**). However, the roles of these particular steps during mammalian oral infection have never been addressed with genetic loss-of-function experiments.

To evaluate the role of NLRC4 inflammasome activation in myeloid cells, we utilized *Nlrc4*⁻/⁻*Rosa26*^LSL-*Nlrc4*^Lyz2^Cre^ mice (here referred to simply as i*Nlrc4*Lyz2^Cre^ mice) (**Rauch et al., 2017**). These mice harbor a germline null mutation in *Nlrc4*, but encode a *Lyz2*^Cre^-inducible *Nlrc4* cDNA transgene

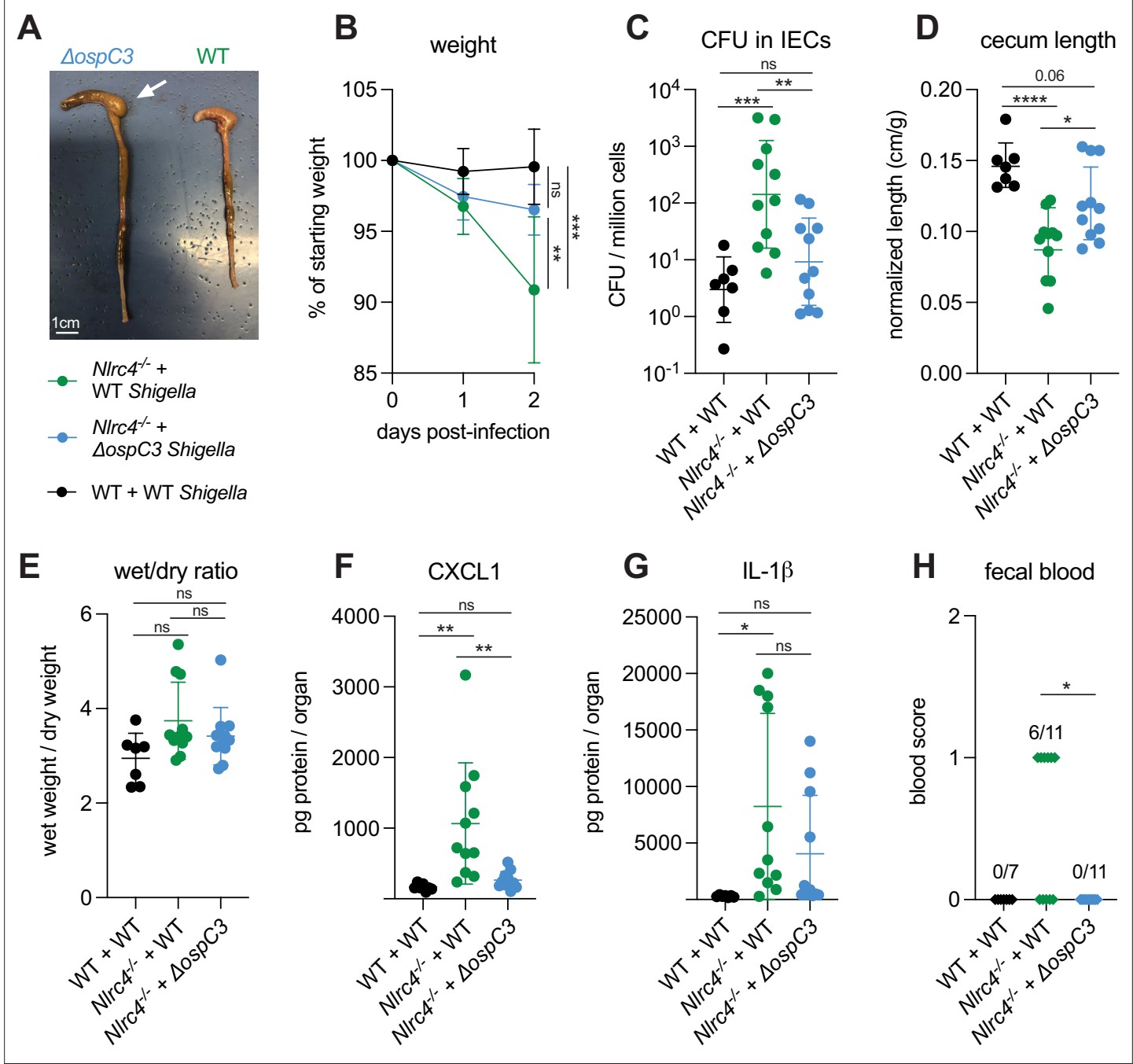

**Figure 3.** *Shigella* effector OspC3 is critical for virulence in oral *Shigella* infection. (**A–H**) Mice were treated orally with 25 mg streptomycin sulfate in water and infected 1 day later. B6.WT mice (co-housed wild-type [WT] and B6.*Nlrc4*+/-*Casp11*+/-) were orally challenged with 10⁷ colony forming units (CFUs) of WT *Shigella flexneri* (n=7) and B6.*Nlrc4*-/- mice (co-housed B6.*Nlrc4*-/- and B6.*Nlrc4*-/-*Casp11*+/-) were challenged with WT (green, n=11) or *ΔospC3 S. flexneri* (blue, n=11). Mice were sacrificed at 2 days post-infection. (**A**) Representative images of the cecum and colon from B6.*Nlrc4*-/- mice infected with WT or *ΔospC3 S. flexneri*. The white arrow indicates clear but reduced inflammation in mice infected with the *ΔospC3* strain. (**B**) Mouse weights from 0 through 2 days post-infection. Each symbol represents the mean for all mice of the indicated genotype. (**C**) *Shigella* CFUs per million cells from the combined intestinal epithelial cell (IEC) enriched fraction of gentamicin-treated cecum and colon tissue. (**D**) Quantification of cecum lengths normalized to mouse weight prior to infection; cecum length (cm)/mouse weight (g). (**E**) The ratio of fecal pellet weight when wet (fresh) divided by the fecal pellet weight after overnight drying. Pellets were collected at day 2 post-infection. (**F, G**) CXCL1 and IL-1β levels measured by ELISA from homogenized cecum and colon tissue of infected mice. (**H**) Blood scores from feces collected at 2 days post-infection. 1=occult blood, 2=macroscopic blood. (**C–H**) Each symbol represents one mouse. Data collected from two independent experiments. Mean ± SD is shown in (**B, D–G**). Geometric mean ± SD is shown in (**C**). Statistical significance was calculated by one-way ANOVA with Tukey's multiple comparison test (**B** (day 2), **C–G**) and by Fisher's

*Figure 3 continued on next page*

Figure 3 continued

exact test in (**H**) where data were stratified by presence (score = 1 or 2) or absence (score = 0) of blood. Data were log-transformed prior to calculations in (**C**) to achieve normality. *p<0.05, **p<0.01, ***p<0.001, ****p<0.0001, ns = not significant (p>0.05).

(integrated within the *Rosa26* locus) that restores NLRC4 expression selectively in myeloid cells (primarily macrophages, monocytes, and neutrophils). We infected WT B6 mice and B6.i*Nlrc4*⁺⁻*Lyz2*^Cre+^ and B6.*Nlrc4*^–/–^ (i*Nlrc4*⁻*Lyz2*^Cre+^) littermates and compared disease outcomes across genotypes (**Figure 5**). Surprisingly, i*Nlrc4*⁺*Lyz2*^Cre+^ mice phenocopied B6.*Nlrc4*^–/–^ mice, and did not exhibit significant differences in weight loss, bacterial colonization of the intestinal epithelium, cecum length, or diarrhea (**Figure 5A–E**). There was a modest but insignificant increase in inflammatory cytokines CXCL1 and IL-1β in B6.*Nlrc4*^–/–^ mice (**Figure 5F and G**), but fewer of these mice displayed fecal blood compared to i*Nlrc4*⁺*Lyz2*^Cre+^ mice (**Figure 5H**). These results provide a striking contrast to our previous results with i*Nlrc4*⁺*VilCre*^Cre+^ mice in which NLRC4 is selectively expressed in IECs (**Mitchell et al., 2020**). Unlike i*Nlrc4*⁺*Lyz2*^Cre+^ mice, i*Nlrc4*⁺*VilCre*^Cre+^ mice were strongly protected from oral *Shigella* infection, implying that epithelial but not myeloid cell NLRC4 is protective. We conclude that NLRC4-dependent pyroptosis in macrophages is neither a major driver of disease pathogenesis nor bacterial colonization in our oral mouse model of infection.

IL-1α and IL-1β are related cytokines that are produced downstream of inflammasome activation in myeloid cells and that signal via the common IL-1 receptor. IL-1 cytokines have been implicated in driving inflammation in the context of mouse intranasal *Shigella* challenge (**Sansonetti et al., 2000**) and rabbit ligated intestinal loop infection (**Sansonetti et al., 1995**). To better address the role of IL-1 in shigellosis, we crossed B6.*Nlrc4*^–/–^ mice to B6.*Il1r1*^–/–^ mice to generate B6.*Nlrc4*^–/–^*Il1r1*^–/–^ double-deficient mice that are susceptible to *Shigella* infection but fail to respond to IL-1. We infected *Nlrc4*⁺*Il1r1*⁺ mice (co-housed B6.WT and *Nlrc4*⁺*Il1r1*⁺/⁻ mice), *Nlrc4*⁺/⁻*Il1r1*^–/–^, *Nlrc4*^–/–^*Il1r1*⁺/⁻, and *Nlrc4*^–/–^*Il1r1*^–/–^ littermates and again assessed disease outcomes (**Figure 6**). Surprisingly, *Nlrc4*^–/–^*Il1r1*^–/–^ mice largely phenocopied *Nlrc4*^–/–^*Il1r1*⁺/⁻ mice. *Nlrc4*^–/–^*Il1r1*^–/–^ appeared less susceptible to weight loss than *Nlrc4*^–/–^*Il1r1*⁺/⁻ mice, although this difference was not statistically significant. Furthermore, we did not observe differences in colonization of the intestinal epithelium, normalized cecum lengths, or inflammatory cytokines (**Figure 6A–E**) between *Nlrc4*^–/–^*Il1r1*^–/–^ and *Nlrc4*^–/–^*Il1r1*⁺/⁻ mice. In many bacterial infections, IL-1 signaling initiates the recruitment of neutrophils to sites of infection. We did not observe a significant difference in the amount of the neutrophil marker myeloperoxidase (MPO) in the feces of *Nlrc4*^–/–^*Il1r1*^–/–^ versus *Nlrc4*^–/–^ *Il1r1*⁺/⁻ mice at 1 day post-infection, however, there was a modest but significant decrease in fecal MPO in *Nlrc4*^–/–^*Il1r1*^–/–^ relative to *Nlrc4*^–/–^ *Il1r1*⁺/⁻ mice at 2 days post-infection, suggesting that IL-1 might be essential for sustained neutrophilic inflammation during *Shigella* infection (**Figure 6F**). We also found that *Nlrc4*⁺/⁻*Il1r1*^–/–^ mice largely phenocopy *Nlrc4*⁺/⁻*Il1r1*⁺/⁻ mice and are resistant to infection. Overall, these results indicate that, despite the increases in IL-1β consistently seen in susceptible mice, IL-1 signaling might affect neutrophil recruitment but is not a primary driver of pathogenesis or protection during oral *Shigella* infection. NLRC4-dependent resistance to shigellosis is therefore likely due to the initiation of pyroptosis and expulsion in IECs and not myeloid cell pyroptosis nor IL-1 signaling. Our results leave open a possible role for another inflammasome-dependent cytokine, IL-18, which unlike IL-1β, is highly expressed in IECs.

## TNFα contributes to resistance to *Shigella*

Given that both NLRC4 and CASP11 protect the mouse epithelium from *Shigella* colonization, we reasoned that additional mechanisms of cell death might function in this niche to counteract *Shigella* invasion and spread. Another cell death initiator in the intestine is TNFα, which has been shown to promote *Salmonella*-induced IEC death and dislodgement (**Fattinger et al., 2021**). TNFα initiates Caspase-8-dependent apoptosis through TNFRI engagement particularly when NF-κB signaling is altered or blocked (**Leppkes et al., 2014**; **Liu et al., 2004**; **Piguet et al., 1998**; **Ruder et al., 2019**). *Shigella* encodes several effectors reported to inhibit NF-κB signaling (**Ashida et al., 2010**; **Ashida et al., 2013**; **de Jong et al., 2016**; **Kim et al., 2005**; **Newton et al., 2010**; **Sanada et al., 2012**; **Wang et al., 2013**), and thus, we hypothesized that TNFα might restrict *Shigella* by inducing death of infected IECs.

To assess the in vivo role of TNFα during shigellosis, we first infected B6.*Nlrc4*^–/–^ mice treated with an antibody that neutralizes TNFα, or with an isotype control antibody (**Figure 7**). B6.*Nlrc4*^–/–^ mice that

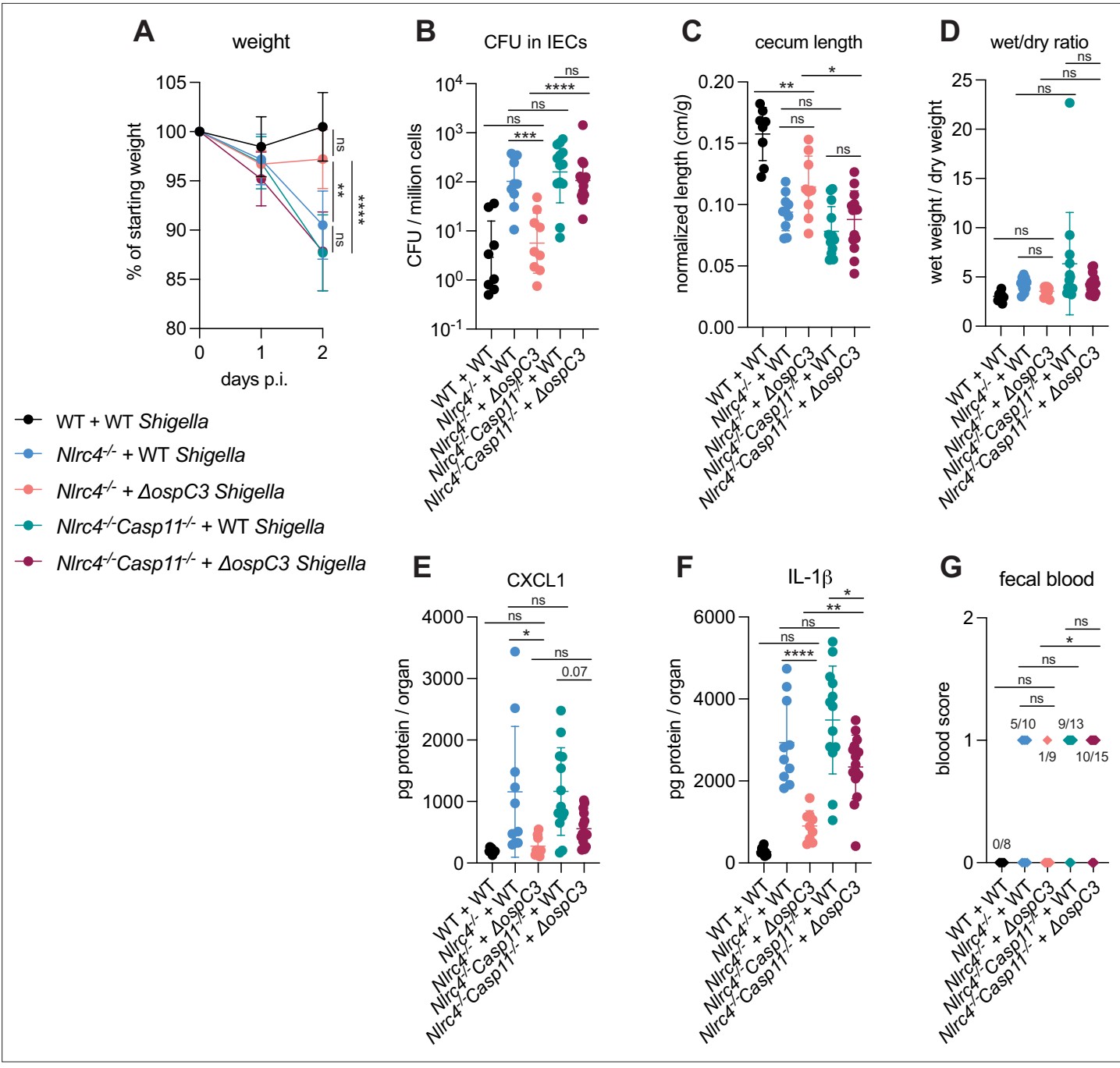

**Figure 4.** OspC3-driven virulence in B6.*Nlrc4*⁻/⁻ mice depends on Caspase-11. (**A–G**) Mice were treated orally with 25 mg streptomycin sulfate in water and then infected 1 day later. B6.WT mice were orally challenged with $10^7$ colony forming units (CFUs) of wild-type (WT) *Shigella flexneri* (black, n=8), B6.*Nlrc4*⁻/⁻ (co-housed B6.*Nlrc4*⁻/⁻*Casp11*⁺/⁺ and B6.*Nlrc4*⁻/⁻*Casp11*⁺/⁻) mice were challenged with WT (blue, n=10) or *ΔospC3 S. flexneri* (pink, n=9), and B6.*Nlrc4*⁻/⁻*Casp11*⁻/⁻ mice (littermates with the B6.*Nlrc4*⁻/⁻*Casp11*⁺/⁻) were challenged with WT (teal, n=13) or *ΔospC3 S. flexneri* (maroon, n=15). Mice were littermates or were co-housed for 3 weeks prior to infection and were sacrificed at 2 days post-infection. (**A**) Mouse weights from 0 through 2 days post-infection. Each symbol represents the mean for all mice of the indicated group. (**B**) *Shigella* CFUs per million cells from the combined intestinal epithelial cell (IEC) enriched fraction of gentamicin-treated cecum and colon tissue. (**C**) Quantification of cecum lengths normalized to mouse weight prior to infection; cecum length (cm)/mouse weight (g). (**D**) The ratio of fecal pellet weight when wet (fresh) divided by the fecal pellet weight after overnight drying. Pellets were collected at day 2 post-infection. (**E, F**) CXCL1 and IL-1β levels measured by ELISA from homogenized cecum and colon tissue of infected mice. (**G**) Blood scores from feces collected at 2 days post-infection. 1=occult blood, 2=macroscopic blood. (**B–G**) Each symbol represents one mouse. Data collected from two independent experiments. Mean ± SD is shown in (**A, C–F**). Geometric mean ± SD is shown in (**B**). Statistical significance was calculated by one-way ANOVA with Tukey's multiple comparison test (**A** (day 2), **B–F**) and by Fisher's exact test in (**G**)

*Figure 4 continued on next page*

*Figure 4 continued*

where data were stratified by presence (score = 1 or 2) or absence (score = 0) of blood. Data were log-transformed prior to calculations in (**B, D**) to achieve normality. *p<0.05, **p<0.01, ***p<0.001, ****p<0.0001, ns = not significant (p>0.05).

underwent TNFα neutralization appeared slightly more susceptible to shigellosis than B6.*Nlrc4⁻/⁻* mice treated with control antibody and displayed trending but insignificant increases in weight loss, bacterial burdens in IECs, IL-1β levels, and fecal blood (**Figure 7A, B, D and E**) and a significant increase in CXCL1 (**Figure 7C**). B6.*Nlrc4⁻/⁻* mice express a functional Caspase-11 inflammasome and given the redundancy we observed between NLRC4 and Caspase-11 (**Figures 1–4**, **Mitchell et al., 2020**), we hypothesized that a protective role for TNFα during *Shigella* infection might be most evident in the absence of both of these cell death pathways. To test this, we repeated the experiment in B6.*Nlrc4⁻/⁻ Casp11⁻/⁻* mice and, indeed, found that TNFα neutralization on this genetic background significantly increased susceptibility to *Shigella* infection. Mice treated with antibody to TNFα experienced an ~5% increase in weight loss, a 10-fold increase in bacterial colonization of the intestinal epithelium, and significant increases in colonic shrinkage, diarrhea, and inflammatory cytokines (**Figure 7F–H and J–N**). There was also a trending but insignificant increase in occult and macroscopic blood in the mice treated with TNFα neutralizing antibody. TNFα levels were elevated significantly in B6.*Nlrc4⁻/⁻ Casp11⁻/⁻* mice, indicating that expression of this cytokine is induced in susceptible mice (**Figure 7N**). The anti-TNFα antibody did not decrease the levels of TNFα measured by ELISA because the antibody neutralizes signaling by the cytokine without interfering with its ability to be detected by ELISA.

Importantly, we could also observe a strong protective role for TNFα in similar experiments performed in 129.*Nlrc4⁻/⁻* mice that are naturally deficient in Caspase-11 (**Figure 7—figure supplement 1**), confirming that TNFα-dependent protection is redundant with both NLRC4 and Caspase-11. These results suggest that a hierarchy of cell death pathways protect the intestinal epithelium from *Shigella* infection. NLRC4 appears to be both necessary and sufficient to protect mice from shigellosis, but in the absence of NLRC4, both Caspase-11 (even in the presence of *Shigella* effector OspC3) and TNFα can provide modest secondary protection (**Figures 1, 2 and 7A–E**). These dual Caspase-11 and TNFα backup pathways appear to have overlapping and compensatory functions during *Shigella* infection, as it is only the removal of both pathways in NLRC4-deficient mice that drives a striking increase in susceptibility to *Shigella* infection. However, since Caspase-11 can significantly but not completely compensate for loss of NLRC4 when *Shigella* lacks OspC3 (**Figure 3** and **Figure 4**) but TNFα appears unable to compensate for loss of NLRC4 (**Figure 7A–E**), Caspase-11 appears to supersede TNFα in the defense hierarchy.

## Loss of multiple cell death pathways renders mice hyper-susceptible to *Shigella*

To test the role of Caspase-8 during *Shigella* infection, we generated mice lacking either Caspases-1 and -11 (B6.*Casp1/11⁻/⁻Casp8⁺/⁻Ripk3⁻/⁻*), Caspase-8 (B6.*Casp1/11⁺/⁻Casp8⁻/⁻Ripk3⁻/⁻*), or Caspases-1, -11, and -8 (B6.*Casp1/11/8⁻/⁻Ripk3⁻/⁻*). Since loss of Caspase-8 results in Ripk3-depedent embryonic lethality, all three genotypes lack *Ripk3*. *Casp1/11⁻/⁻Casp8⁺/⁻Ripk3⁻/⁻* mice retain Caspase-8 function downstream of both NLRC4 and TNFα (**Figure 2—figure supplement 2**) and based on our previous experiments with *Casp1/11⁻/⁻* mice (**Mitchell et al., 2020**), we expected that these mice would be resistant to infection. Similarly, *Casp1/11⁺/⁻Casp8⁻/⁻Ripk3⁻/⁻* mice retain the ability to recruit Caspase-1 to NLRC4 and to initiate cell death via Caspase-11 (**Figure 2—figure supplement 2**) and should also thus be resistant to infection. *Casp1/11/8⁻/⁻Ripk3⁻/⁻* mice, however, should lack the cell death pathways initiated by NLRC4 (via Caspase-1 or Caspase-8), Caspase-11, and TNFα (**Figure 2—figure supplement 2**), and our results above suggest that these mice might be highly susceptible to infection.

We infected WT B6 mice and *Casp1/11⁻/⁻Casp8⁺/⁻Ripk3⁻/⁻*, *Casp1/11⁺/⁻Casp8⁻/⁻Ripk3⁻/⁻*, and *Casp1/11/8⁻/⁻Ripk3⁻/⁻* littermates that had been co-housed with the WT mice and assessed disease phenotypes across all four genotypes (**Figure 8**). We found that *Casp1/11⁺/⁻Casp8⁻/⁻Ripk3⁻/⁻* mice largely phenocopied WT B6 mice, and were resistant to infection, exhibiting minimal weight loss, diarrhea, cecal or colonic shrinkage, and no fecal blood (**Figure 8A, B, D, E, F, I**). Furthermore, we could not detect significant increases in bacterial burdens in the intestinal epithelium (**Figure 8C**) nor inflammatory cytokines (**Figure 8G and H**) in *Casp1/11⁺/⁻Casp8⁻/⁻Ripk3⁻/⁻* mice. These results suggest that Caspase-8 and RIPK3 are not necessary for resistance to *Shigella* in the presence of functional

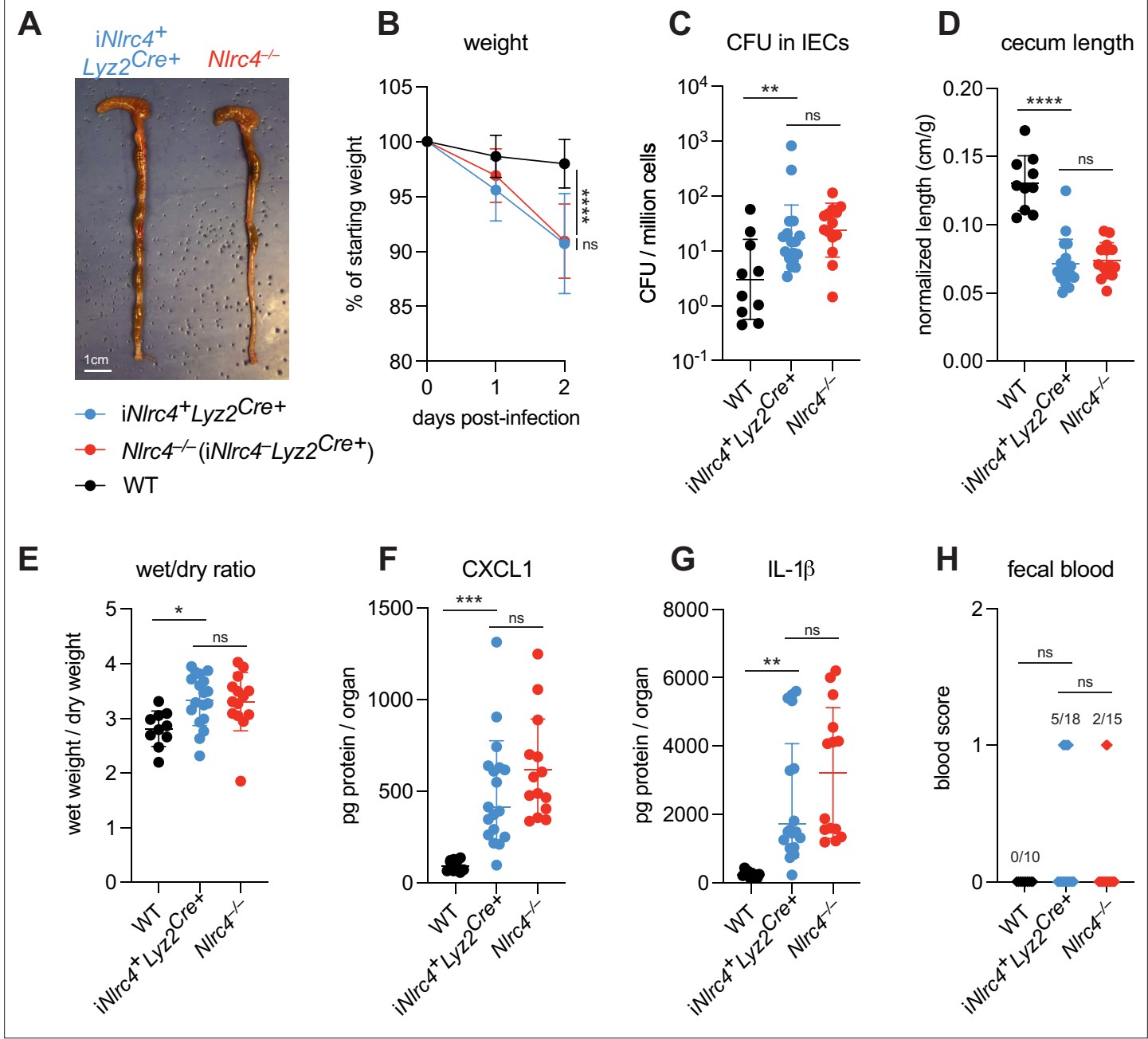

**Figure 5.** NLRC4 in myeloid-derived cells does not affect *Shigella* pathogenesis. (**A–H**) B6.WT (black, n=10) mice were co-housed with B6.i*Nlrc4*+*Lyz2*^Cre+ (blue, n=18) and B6.*Nlrc4*^–/– (i*Nlrc4*–*Lyz2*^Cre+, red, n=15) littermates, treated orally with 25 mg streptomycin sulfate in water, and orally challenged the next day with $10^7$ colony forming units (CFUs) of wild-type (WT) *Shigella flexneri*. Mice were sacrificed at 2 days post-infection. (**A**) Representative images of the cecum and colon from i*Nlrc4*+*Lyz2*^Cre+ and B6.*Nlrc4*^–/– mice. Note the similarity in gross pathology between the two genotypes. (**B**) Mouse weights from 0 through 2 days post-infection. Each symbol represents the mean for all mice of the indicated group. (**C**) *Shigella* CFUs per million cells from the combined intestinal epithelial cell (IEC) enriched fraction of gentamicin-treated cecum and colon tissue. (**D**) Quantification of cecum lengths normalized to mouse weight prior to infection; cecum length (cm)/mouse weight (g). (**E**) The ratio of fecal pellet weight when wet (fresh) divided by the fecal pellet weight after overnight drying. Pellets were collected at day 2 post-infection. (**F, G**) CXCL1 and IL-1β levels measured by ELISA from homogenized cecum and colon tissue of infected mice. (**H**) Blood scores from feces collected at 2 days post-infection. 1=occult blood, 2=macroscopic blood. (**C–H**) Each symbol represents one mouse. Data collected from two independent experiments. Mean ± SD is shown in (**B, D–G**). Geometric mean ± SD is shown in (**C**). Statistical significance was calculated by one-way ANOVA with Tukey's multiple comparison test (**B** (day 2), **C–G**) and by Fisher's exact test in (**H**) where data were stratified by presence (score = 1 or 2) or absence (score = 0) of blood. Data were log-transformed prior to calculations in (**C**) to achieve normality. *p<0.05, **p<0.01, ***p<0.001, ****p<0.0001, ns = not significant (p>0.05).

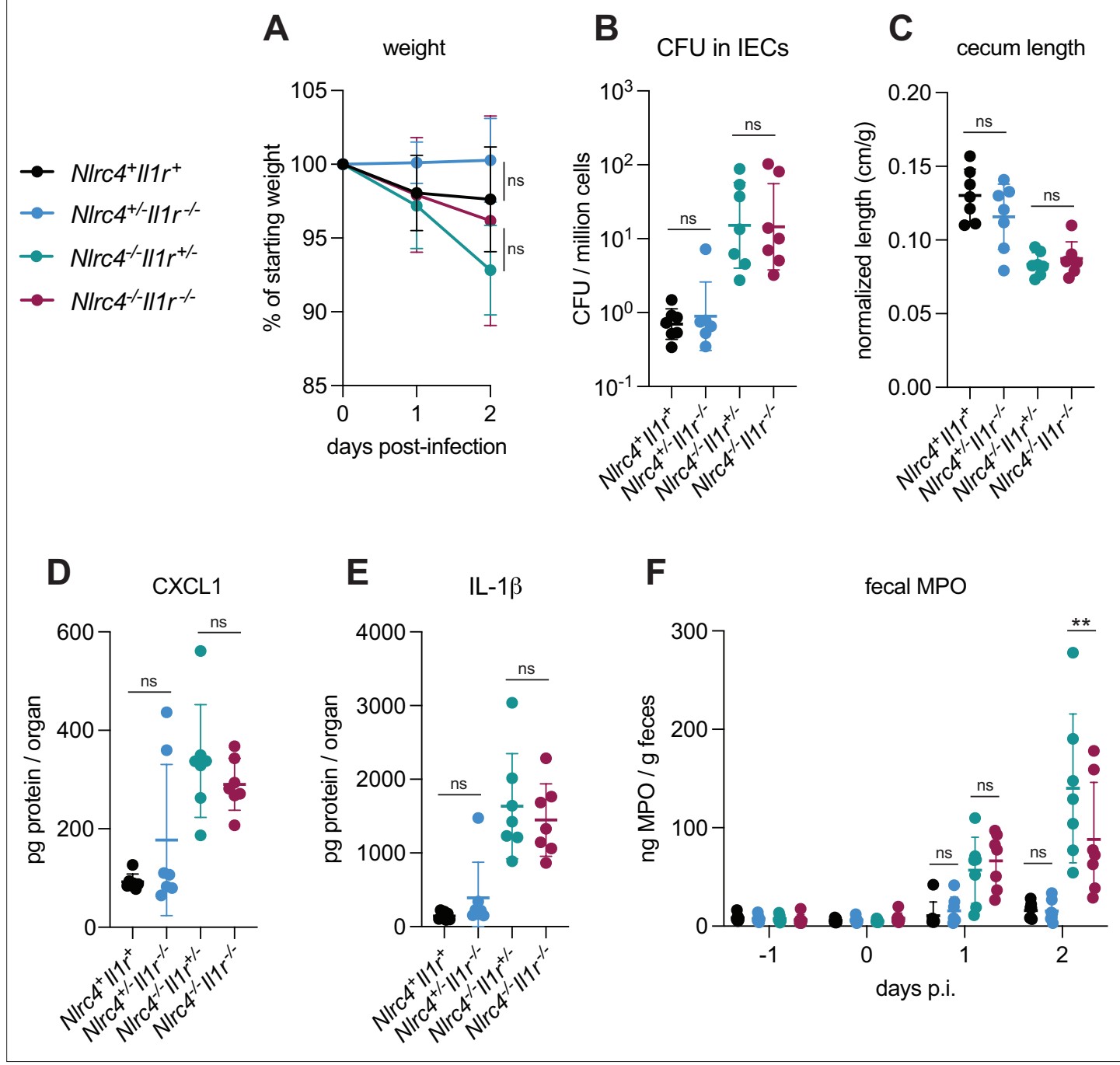

**Figure 6.** IL-1 signaling does not affect *Shigella* pathogenesis. (**A–F**) *Nlrc4+Il1r1+*mice (co-housed B6.WT and *Nlrc4+/–Il1r1+/–*, black, n=7), *Nlrc4+/–Il1r1–/–* (blue, n=7), *Nlrc4–/–Il1r1+/–* (teal, n=7), and *Nlrc4–/–Il1r1–/–* (maroon, n=7) littermates were treated orally with 25 mg streptomycin sulfate in water and orally challenged the next day with $10^7$ colony forming units (CFUs) of wild-type (WT) *Shigella flexneri*. Mice were sacrificed at 2 days post-infection. (**A**) Mouse weights from 0 through 2 days post-infection. Each symbol represents the mean for all mice of the indicated group. (**B**) *Shigella* CFUs per million cells from the combined intestinal epithelial cell (IEC) enriched fraction of gentamicin-treated cecum and colon tissue. (**C**) Quantification of cecum lengths normalized to mouse weight prior to infection; cecum length (cm)/mouse weight (g). (**D, E**) CXCL1 and IL-1β levels measured by ELISA from homogenized cecum and colon tissue of infected mice. (**F**) Myeloperoxidase enzyme levels in mouse feces collected each day prior to and during infection and measured by ELISA. (**B–F**) Each symbol represents one mouse. Data were collected from one experiment but are representative of two independent experiments. Mean ± SD is shown in (**A, C–F**). Geometric mean ± SD is shown in (**B**). Statistical significance was calculated by one-way ANOVA with Tukey's multiple comparison test (**A** day 2), **B–E**) and two-way ANOVA with Tukey's multiple comparison test (**F**). Data were log-transformed prior to calculations in (**B**) to achieve normality. *p<0.05, **p<0.01, ***p<0.001, ****p<0.0001, ns = not significant (p>0.05).

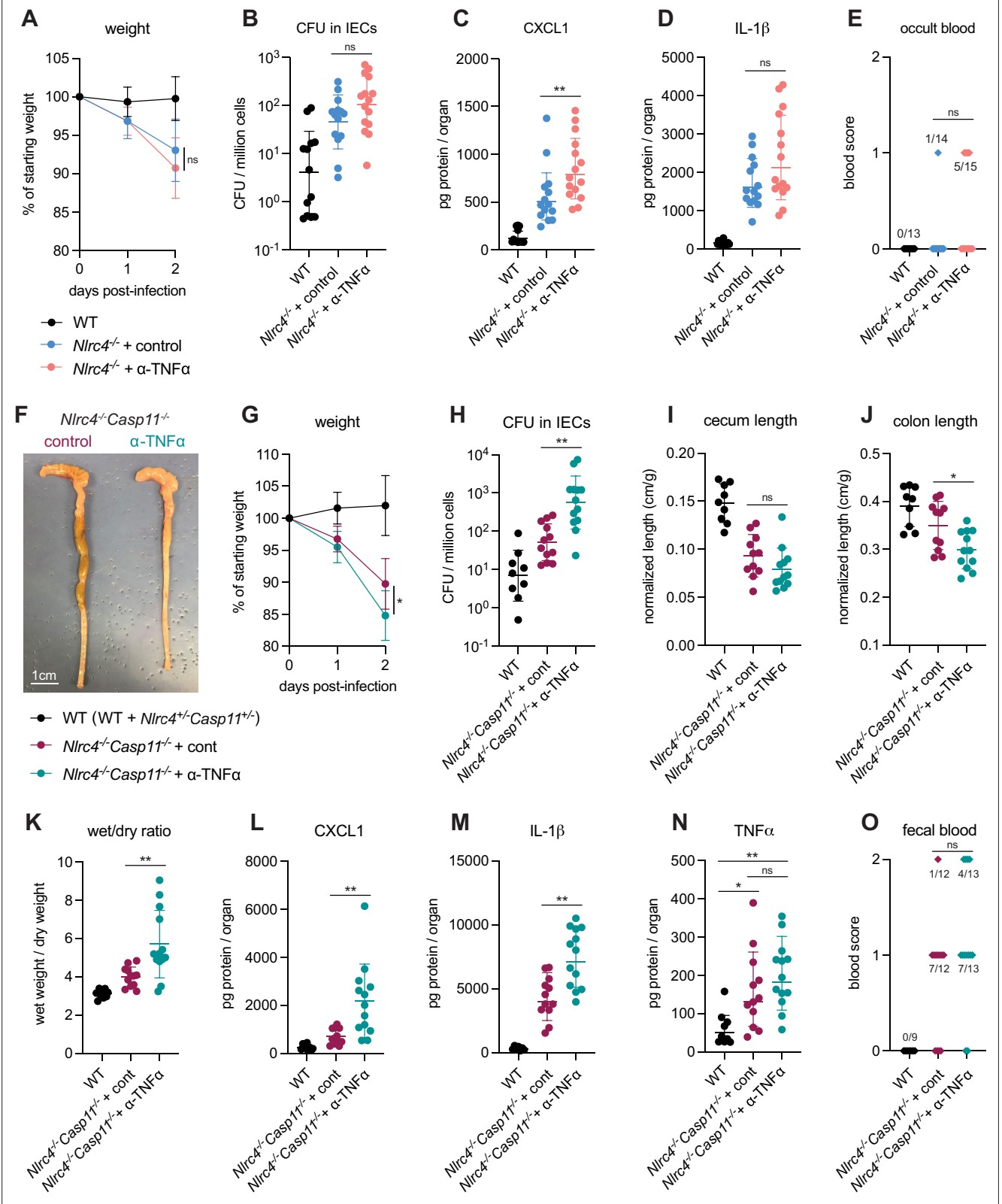

**Figure 7.** TNFα contributes to resistance to *Shigella* when mice lack NLRC4 and CASP11. Wild-type (WT) (B6.WT, black, n=13 for (**A–E**), and both co-housed B6.WT and B6.*Nlrc4+/−Casp11+/−*, black, n=9 for (**F–O**)), B6.*Nlrc4−/−*, and B6.*Nlrc4−/−Casp11−/−* mice were treated orally with 25 mg streptomycin sulfate in water and orally challenged the next day with 10^7 colony forming units (CFUs) of WT *Shigella flexneri*. In (**A–E**), B6.*Nlrc4−/−* mice received 200 µg of either TNFα neutralizing antibody (pink, n=13) or isotype control antibody (light blue, n=14) by intraperitoneal injection daily from 1 day before

*Figure 7 continued on next page*

*Figure 7 continued*
infection through sacrifice at 2 days post-infection. In (**F–O**), B6.*Nlrc4*⁻/⁻*Casp11*⁻/⁻ mice received 200 µg of either TNFα neutralizing antibody (teal, n=12) or isotype control antibody (maroon, n=13) by intraperitoneal injection daily from 1 day before infection through sacrifice at 2 days post-infection. (**A, G**) Mouse weights from 0 through 2 days post-infection. Each symbol represents the mean for all mice of the indicated group. (**B, H**) *Shigella* CFUs per million cells from the combined intestinal epithelial cell (IEC) enriched fraction of gentamicin-treated cecum and colon tissue. (**C, D, L–N**) CXCL1, IL-1β, and TNFα levels measured by ELISA from homogenized cecum and colon tissue of infected mice. (**E, O**) Blood scores from feces collected at 2 days post-infection. 1=occult blood, 2=macroscopic blood. (**F**) Representative images of the cecum and colon from B6.*Nlrc4*⁻/⁻*Casp11*⁻/⁻ mice receiving either isotype control or TNFα neutralizing antibody. (**I, J**) Quantification of cecum and colon lengths normalized to mouse weight prior to infection; cecum or colon length (cm)/mouse weight (g). (**K**) The ratio of fecal pellet weight when wet (fresh) divided by the fecal pellet weight after overnight drying. Pellets were collected at day 2 post-infection. (**B–E, H–O**) Each symbol represents one mouse. Data collected from three independent experiments (**A–E**) and two independent experiments (**F–O**). Mean ± SD is shown in (**A, C, D, G, I–N**). Geometric mean ± SD is shown in (**B, H**). Statistical significance was calculated by Mann-Whitney test in (**A** (day 2), **B–D**, **G** (day 2), **H–M**), by one-way ANOVA with Tukey's multiple comparison test in (**N**), and by Fisher's exact test in (**E, O**) where data were stratified by presence (score = 1 or 2) or absence (score = 0) of blood. *p<0.05, **p<0.01, ***p<0.001, ****p<0.0001, ns = not significant (p>0.05).

The online version of this article includes the following figure supplement(s) for figure 7:

**Figure supplement 1.** TNFα neutralization renders 129.*Nlrc4*⁻/⁻ mice more susceptible to *Shigella*.

NLRC4–CASP1 and CASP11 inflammasomes. Interestingly, *Casp1/11*⁻/⁻*Casp8*⁺/⁻*Ripk3*⁻/⁻ mice were not fully resistant to disease and experienced modest but significant weight loss (~5% relative to WT) and significant increases in cecal and colonic shrinkage (*Figure 8B, E and F*). These mice also exhibited trending but insignificant increases in diarrhea, inflammatory cytokines CXCL1 and IL-1β, and fecal blood (*Figure 8D, G, H, I*). This result indicates that Caspase-8 is not sufficient to render mice fully resistant to *Shigella* infection, perhaps because it is antagonized by *Shigella* effector OspC1, which suppresses Caspase-8 activity in human cell lines (*Ashida et al., 2020*).

The most striking observation was that *Casp1/11/8*⁻/⁻*Ripk3*⁻/⁻ mice were highly susceptible to *Shigella* infection, exhibiting severe weight loss (~15% of starting weight), diarrhea, and cecal and colonic shrinkage (*Figure 8A, B and D–F*). These mice also exhibited a massive (>500×) increase in bacterial colonization of the intestinal epithelium (*Figure 8C*) and elevated levels of inflammatory cytokines (*Figure 8G and H*). All *Casp1/11/8*⁻/⁻*Ripk3*⁻/⁻ mice presented with blood in their feces (*Figure 8I*) and one of the ten mice also died of shigellosis within 2 days of infection. The ceca and colons of *Casp1/11/8*⁻/⁻*Ripk3*⁻/⁻ mice were highly inflamed – the tissue thickened, turned white, and sections of the epithelium appeared to have been shed into the lumen, which was completely devoid of feces and filled instead with neutrophilic pus (*Figure 8A*). While the most significant inflammation in B6.*Nlrc4*⁻/⁻ mice is typically seen in the cecum (*Mitchell et al., 2020*), we noted that the colon of *Casp1/11/8*⁻/⁻*Ripk3*⁻/⁻ mice was highly inflamed as well (*Figure 8A and F*), suggesting that a protective role for Caspase-8 might be most important in this organ. The striking difference in susceptibility between *Casp1/11*⁻/⁻*Casp8*⁺/⁻*Ripk3*⁻/⁻ and *Casp1/11/8*⁻/⁻*Ripk3*⁻/⁻ suggests that any inhibition of Caspase-8 by OspC1, if present, is modest. Indeed, the activity of this effector might be specific to human cells.

Taken together, our results imply that redundant cell death pathways protect mice from disease upon oral *Shigella* challenge. Genetic removal of three caspases essential to this response leads to severe shigellosis. However, removal of one or two caspases critical to this response does not lead to severe disease because of significant compensation from the other pathway(s). We observe a hierarchical importance of the cell death pathways, namely, NLRC4>CASP11>TNFα–CASP8 (*Figure 2—figure supplement 2*). We speculate that this hierarchy may be established by the order in which a pathway can sense invasive *Shigella* within the epithelium and initiate a cell death response.

## Discussion

We have previously shown that IEC expression of the NAIP–NLRC4 inflammasome is sufficient to confer resistance to shigellosis in mice (*Mitchell et al., 2020*). Activation of NAIP–NLRC4 by *Shigella* drives pyroptosis and expulsion of infected IECs. Genetic removal of NAIP–NLRC4 from IECs allows *Shigella* to colonize the intestinal epithelium, an event which drives intestinal inflammation and disease. Mouse IECs, however, deploy additional initiators of programmed cell death (*Patankar and Becker, 2020*) and it remained an open question whether these cell death pathways might also counteract *Shigella*.

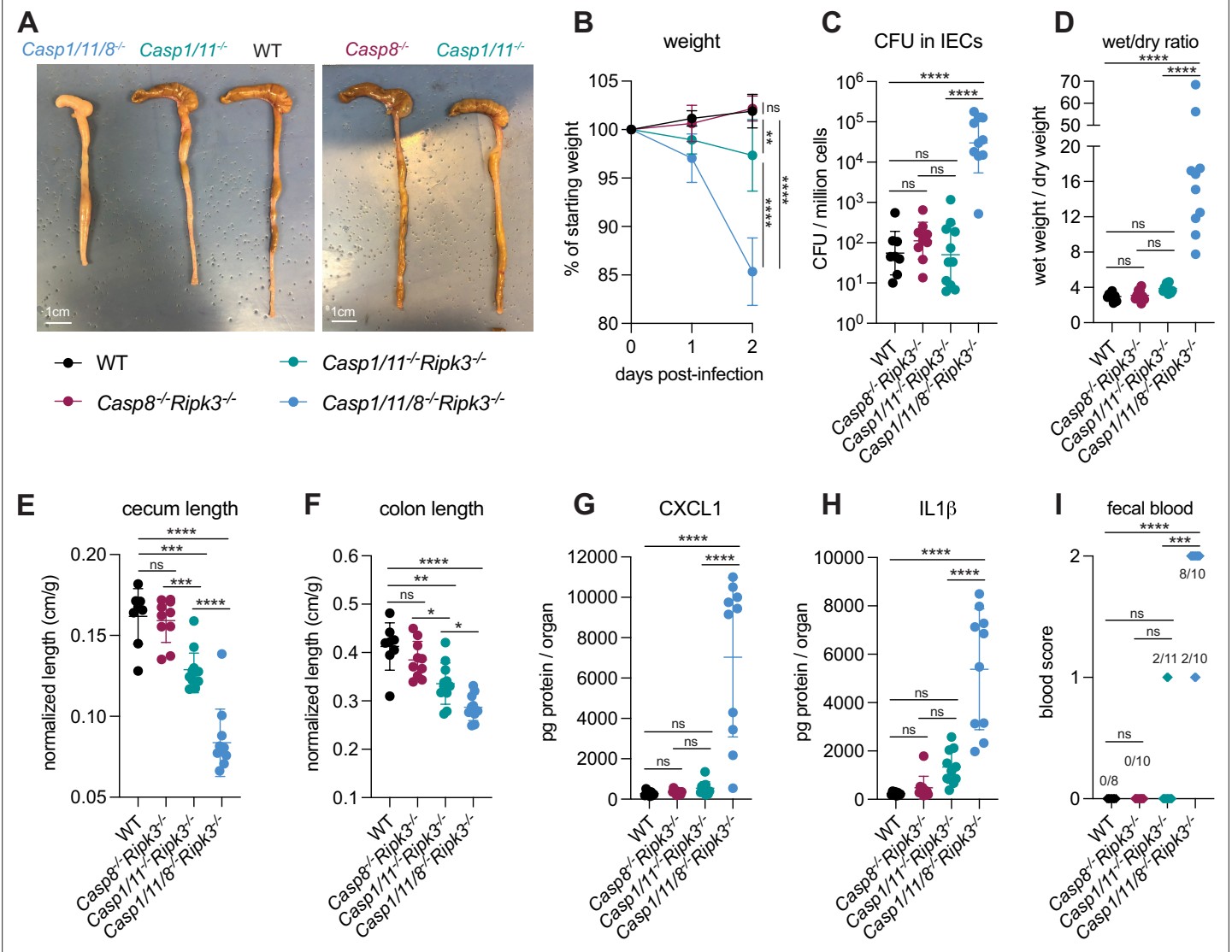

**Figure 8.** Loss of multiple cell death pathways renders mice hyper-susceptible to *Shigella*. (**A–I**) B6.WT mice (black, n=8) were co-housed with B6.*Casp8⁻/⁻Ripk3⁻/⁻* (B6. *Casp1/11⁺/⁻Casp8⁻/⁻Ripk3⁻/⁻*, maroon, n=10), B6.*Casp1/11⁻/⁻Ripk3⁻/⁻* (B6.*Casp1/11⁻/⁻Casp8⁺/⁻Ripk3*, teal, n=11), and B6.*Casp1/11/8⁻/⁻Ripk3⁻/⁻* (light blue, n=10) littermates and treated orally with 25 mg streptomycin sulfate in water and orally challenged the next day with 10⁷ colony forming units (CFUs) of wild-type (WT) *Shigella flexneri*. Mice were sacrificed at 2 days post-infection. (**A**) Representative images of the cecum and colon of infected B6.WT, B6.*Casp8⁻/⁻Ripk3⁻/⁻*, B6.*Casp1/11⁻/⁻Ripk3⁻/⁻*, and *Casp1/11/8⁻/⁻Ripk3⁻/⁻* mice. Note the severe inflammation in the *Casp1/11/8⁻/⁻Ripk3⁻/⁻* mice (left-most organs). (**B**) Mouse weights from 0 through 2 days post-infection. Each symbol represents the mean for all mice of the indicated group. (**C**) *Shigella* CFUs per million cells from the combined intestinal epithelial cell (IEC) enriched fraction of gentamicin-treated cecum and colon tissue. (**D**) The ratio of fecal pellet weight when wet (fresh) divided by the fecal pellet weight after overnight drying. Pellets were collected at day 2 post-infection. (**E, F**) Quantification of cecum and colon lengths normalized to mouse weight prior to infection; cecum or colon length (cm)/mouse weight (g). (**G, H**) CXCL1 and IL-1β levels measured by ELISA from homogenized cecum and colon tissue of infected mice. (**I**) Blood scores from feces collected at 2 days post-infection. 1=occult blood, 2=macroscopic blood. (**C–I**) Each symbol represents one mouse. Data collected from two independent experiments. Mean ± SD is shown in (**B, D–H**). Geometric mean ± SD is shown in (**C**). Statistical significance was calculated by one-way ANOVA with Tukey's multiple comparison test (**B** (day 2), **C–H**) and by Fisher's exact test in (**I**) where data were stratified by presence (score = 1 or 2) or absence (score = 0) of blood. Data were log-transformed prior to calculations in (**C, D**) to achieve normality. *p<0.05, **p<0.01, ***p<0.001, ****p<0.0001, ns = not significant (p>0.05).

We utilized the natural variation in 129.*Nlrc4⁻/⁻* mice, which lack functional CASP11 (*Kayagaki et al., 2011*), to show that CASP11 partially controls the difference in susceptibility between 129.*Nlrc4⁻/⁻* and B6.*Nlrc4⁻/⁻* mice (*Figure 1, Figure 1—figure supplement 1*). In F₁ 129/B6.*Nlrc4⁻/⁻*×129.*Nlrc4⁻/⁻* back-crossed mice, which were either 129/129 or B6/129 at the *Casp11* locus, increased disease severity

and colonization of the intestinal epithelium was associated with a homozygous null *Casp11*[129] locus. We also investigated the role of *Hiccs*, a locus present in 129 mice that confers increased susceptibility to *H. hepaticus*-induced colitis (*Boulard et al., 2012*). The 129 *Hiccs* locus contains polymorphisms in the *Alpk1* gene which encodes alpha-kinase 1 (ALPK1), an activator of NF-κB which has been shown to sense *Shigella*-derived ADP-heptose in human cells (*Zhou et al., 2018*). However, we did not find evidence that the natural variation in *Hiccs* in 129 versus B6 mice contributed to differences in susceptibility between the two strains (*Figure 1—figure supplement 2*).

We observed that *ΔospC3 Shigella* is significantly attenuated in B6.*Nlrc4*[−/−] mice but not in B6.*Nlrc4*[−/−]*Casp11*[−/−], indicating by a 'genetics squared' analysis (*Persson and Vance, 2007*) that *Shigella* effector OspC3 inhibits CASP11 during oral mouse infection (*Figures 3 and 4*). The striking decrease in colonization of the intestinal epithelium in *ΔospC3*-infected B6.*Nlrc4*[−/−] mice relative to *ΔospC3*-infected B6.*Nlrc4*[−/−]*Casp11*[−/−] mice suggests that CASP11-dependent protection is epithelial-intrinsic. *Shigella* also deploys an effector, IpaH7.8, which degrades human (but not mouse) GSDMD to block pyroptosis, further underscoring the importance of this axis in defense (*Luchetti et al., 2021*). We note that CASP11-dependent protection is not sufficient to render *ΔospC3*-infected B6.*Nlrc4*[−/−] mice fully resistant to disease symptoms, perhaps because the priming required to induce CASP11 expression might delay its protective response (*Oh et al., 2021*).

Despite its role as a key cell death initiator in the gut (*Patankar and Becker, 2020*; *Piguet et al., 1998*; *Ruder et al., 2019*), TNFα has not yet been shown to play a major role in defense against pathogens that colonize the intestinal epithelium. Indeed, its role is usually reported to be detrimental to the host. For example, TNFα is a major driver of pathology during Crohn's disease (*van Dullemen et al., 1995*). In the context of *Salmonella* infection, TNFα appears to drive widespread pathological death and dislodgement of IECs at 72 hr post-infection (*Fattinger et al., 2021*). Here, we show that TNFα is protective during oral *Shigella* infection, providing a rationale for why this cytokine is produced in the intestine. In both B6.*Nlrc4*[−/−]*Casp11*[−/−] and 129.*Nlrc4*[−/−] mice, TNFα neutralization led to a striking increase in severity of infection and a 10-fold increase in bacterial colonization of the intestinal epithelium (*Figure 7*, *Figure 7—figure supplement 1*).

TNFα-dependent protection might occur via an NF-κB-dependent, pro-inflammatory response from infected or bystander IECs that express TNFRI or by TNFRI-CASP8-dependent apoptosis of infected cells. Given the redundant, overlapping functions of both Caspase-11 and TNFα in the absence of NLRC4, we favor the hypothesis that TNFα promotes epithelial defense by initiating IEC apoptosis of infected cells in which NF-κB signaling is blocked (*Ashida et al., 2010*; *Ashida et al., 2013*; *de Jong et al., 2016*; *Kim et al., 2005*; *Newton et al., 2010*; *Sanada et al., 2012*; *Wang et al., 2013*). NF-κB-dependent cytokines IL-1β and CXCL1 increase after TNFα neutralization, hinting that protection might not be driven by the TNFα-dependent activation of NF-κB. However, this interpretation is complicated by the fact that bacterial burdens also increase and might drive the observed increases in NF-κB-dependent cytokines via an alternate mechanism. An important next step will be to associate TNFα-dependent protection with expulsion of infected IECs in the mouse gut or in IEC organoid cultures. Co-staining for cleaved Caspase-8 in these experiments would further support our hypothesis that TNFα promotes clearance of *Shigella* via extrinsic apoptosis. Identification of *Shigella* effectors (*Ashida et al., 2010*; *Ashida et al., 2013*; *de Jong et al., 2016*; *Kim et al., 2005*; *Newton et al., 2010*; *Sanada et al., 2012*; *Wang et al., 2013*) that block mouse NF-κB signaling and promote apoptosis of infected cells in vivo is the subject of ongoing investigation. Indeed, existing reports that *Shigella* suppresses CASP8–dependent apoptosis in human epithelial cells further implicate this cell death pathway in defense (*Ashida et al., 2020*; *Faherty et al., 2010*).

We find that *Casp1/11/8*[−/−]*Ripk3*[−/−] mice, which lack the pathways to execute pyroptosis, extrinsic apoptosis, and necroptosis, experience severe shigellosis with a 500-fold increase in colonization of the intestinal epithelium relative to B6 WT mice (*Figure 8*). Although we did not directly compare the two mouse strains, *Casp1/11/8*[−/−]*Ripk3*[−/−] mice (*Figure 8*) experienced more severe disease and epithelial colonization than *Nlrc4*[−/−]*Casp11*[−/−] mice (*Figures 2, 4 and 7*). We speculate that the additional susceptibility of *Casp1/11/8*[−/−]*Ripk3*[−/−] mice results from the absence of TNFRI–CASP8-dependent apoptosis and possibly from the absence of RIPK3-dependent necroptosis. While both apoptosis and necroptosis appear to be blocked in human cells by *Shigella* effectors OspC1 and OspD3, respectively (*Ashida et al., 2020*), the critical protective role of Caspase-8 in the absence of Caspase-1, Caspase-11, and RIPK3 suggests that this cell death initiator is not strongly antagonized by

OspC1 in mice. Robust CASP8-dependent activity might intrinsically prevent necroptosis (*Jorgensen et al., 2017*; *Wen et al., 2017*), thus rendering OspD3 unimportant in the context of mouse *Shigella* infection, regardless of its ability to target mouse RIPK1 and RIPK3. We do observe that *Casp1/11⁻/⁻ Casp8⁺/⁻Ripk3⁻/⁻* are modestly susceptible to infection while *Casp1/11⁺/⁻Casp8⁻/⁻Ripk3⁻/⁻* mice are fully resistant. This difference might be the result of a modest and incomplete CASP8 blockade by OspC1, as described above, or because NLRC4–CASP8-dependent cell death is delayed relative to NLRC4–CASP1-dependent cell death (*Lee et al., 2018*; *Rauch et al., 2017*). In addition, we note that CASP8 is a pleiotropic enzyme and might contribute to defense against *Shigella* via a mechanism that is independent of TNFα or NLRC4-dependent cell death (*Gitlin et al., 2020*; *Philip et al., 2016*; *Schwarzer et al., 2020*; *Stolzer et al., 2022*; *Weng et al., 2014*; *Woznicki et al., 2021*).

Despite the commonly held belief that macrophage pyroptosis and IL-1 signaling drive *Shigella* pathogenesis (*Schnupf and Sansonetti, 2019*; *Schroeder and Hilbi, 2008*), we find no major protective or pathogenic role for either during *Shigella* infection (*Figures 5 and 6*). These data suggest that epithelial-specific cell death and expulsion may be the key mechanism that protects mice from *Shigella*. Infections in IL-18-deficient mice will further clarify the role of inflammasome-dependent cytokines in protection. Additional studies in bone marrow chimeric mice or tissue-specific knockout mice are required to genetically confirm whether the protective effects of CASP11 and TNFα are epithelial-intrinsic. While we infer that cell death in the intestinal epithelium is the protective mechanism downstream of both CASP11 and TNFα, further experiments are required to directly observe and quantify differences in these modes of cell death in vivo.

Taken together, our experiments suggest the existence of a layered cell death pathway hierarchy (NLRC4>CASP11>TNFα–CASP8) that is essential in defense against oral *Shigella* infection in mice. Our work highlights both the importance of redundant layers of immunity as a strategy to counteract intracellular pathogens and the significant evolutionary steps required by *Shigella* to overcome these pathways and cause disease in humans. We observed a correlation between bacterial burdens in IECs and pathogenicity in our experiments, indicating that the extent to which *Shigella* can colonize the intestinal epithelium dictates the severity of disease during infection. However, the sensors within IECs that initiate inflammation and drive pathogenicity in vivo have yet to be uncovered and might present an ideal pharmacological target to limit pathological inflammation during acute *Shigella* infection.

## Materials and methods

### Key resources table

| Reagent type (species) or resource | Designation | Source or reference | Identifiers | Additional information |
|---|---|---|---|---|
| Strain, strain background (*Mus musculus*, C57BL/6J) | WT | Jax and Vance Lab colony, Jax stock No. 000664 | | |
| Strain, strain background (*Mus musculus*, C57BL/6J) | *Nlrc4⁻/⁻* | Vance Lab colony *Tenthorey et al., 2020* | | Crossed to 129. *Nlrc4⁻/⁻* mice for mapping studies |
| Strain, strain background (*Mus musculus*, C57BL/6J) | *Casp11⁻/⁻* | Vance Lab colony, this paper | | |
| Strain, strain background (*Mus musculus*, C57BL/6J) | *Il1r1⁻/⁻* | Jax and Vance Lab colony, Jax stock No. 003245 | | |
| Strain, strain background (*Mus musculus*, C57BL/6J) | *Casp1/11/8⁻/⁻ Ripk3⁻/⁻* | Vance Lab colony *Rauch et al., 2017* | | |
| Strain, strain background (*Mus musculus*, C57BL/6J and C57BL/6N mixed) | *Rosa26^{LSL-Nlrc4}* (formerly called i*Nlrc4*) | Vance Lab colony *Rauch et al., 2017* | | Encode a *Cre*-inducible *Nlrc4* gene in the *Rosa26* locus |

*Continued on next page*

*Continued*

| Reagent type (species) or resource | Designation | Source or reference | Identifiers | Additional information |
|---|---|---|---|---|
| Strain, strain background (*Mus musculus*, C57BL/6J) *Lyz2^Cre* | | Jax and Vance Lab Colony, Jax stock No. 004781 | | |
| Strain, strain background (*Mus musculus*, 129S1/SvImJ) | WT | Jax and Vance Lab colony, Jax stock No. 002448 | | |
| Strain, strain background (*Mus musculus*, 129S1/SvImJ) | *Nlrc4^−/−^* | Vance Lab colony **Mitchell et al., 2020** | | Crossed to B6. *Nlrc4^−/−^* mice for mapping studies |
| Strain, strain background (*Shigella flexneri* serovar 2a) | WT 2457T | Lesser Lab | | Streptomycin resistant |
| Strain, strain background (*Shigella flexneri* serovar 2a) | *ΔospC3* 2457T | Lesser Lab **Mou et al., 2018** | | Streptomycin resistant |
| Antibody | Rat anti-mIL-1β capture and goat anti-mIL-1β polyclonal detection antibodies | R&D | DY401 | For ELISA (each used at 100 µL per well) |
| Antibody | Rat anti-mCXCL1 capture and rat anti-mCXCL1 detection antibodies | R&D | DY453 | For ELISA (each used at 100 µL per well) |
| Antibody | Goat anti-mMPO capture and goat anti-mMPO detection antibodies | R&D | DY3667 | For ELISA (each used at 100 µL per well) |
| Antibody | Monoclonal anti-TNFα capture and detection antibodies | Thermo Fisher | BMS607HS | For ELISA. Capture antibody is precoated on purchased plates, detection antibody used at 50 µL per well |
| Antibody | Hamster anti-TNFα monoclonal neutralizing antibody | Bio X cell | TN3-19.12 | In vivo treatments, 200 µg daily |
| Antibody | Polyclonal Armenian hamster IgG isotype control | Bio X cell | BE0091 | In vivo treatments,, 200 µg daily |
| Antibody | Rat anti-mCasp11 monoclonal antibody | Novus | 17D9 | 1:500 |

## Animal procedures

All mice were maintained in a specific pathogen-free colony until 1–8 weeks prior to infection, maintained under a 12 hr light-dark cycle (7 am to 7 pm), and given a standard chow diet (Harlan irradiated laboratory animal diet) ad libitum. Animals used in infection experiments were littermates or, if not possible, were generally co-housed upon weaning. In cases when mice were not co-housed upon weaning, mice were co-housed for at least 3 weeks prior to infection. Co-housing was strategically performed to maximize cage overlap between all experimental groups. Different experimental treatments (comparing disease across different *Shigella* genotypes or antibody treatments) were stratified within mouse genotypes of the same litter, where possible, to ensure that phenotypes were not the result of the differences in different litter microbiomes. Mice were transferred from an SPF colony to an ABSL2 facility at least 1 week prior to infection. All mouse infections complied with the regulatory standards of, and were approved by, the University of California, Berkeley Animal Care and Use Committee. B6.*Nlrc4^−/−^* (C57BL/6J background) and 129.*Nlrc4^−/−^* (129S1/SvImJ background) mice were

generated as previously described (**Mitchell et al., 2020**; **Tenthorey et al., 2020**). $F_1$ 129/B6.$Nlrc4^{-/-}$ were generated by crossing parental 129.$Nlrc4^{-/-}$ and B6.$Nlrc4^{-/-}$ mice. $F_1$ 129/B6.$Nlrc4^{-/-}$ mice were crossed to parental 129.$Nlrc4^{-/-}$ mice to generate backcrossed mice that were either B6/129 or 129/129 at each locus. 129 and B6 *Casp11* alleles were distinguished by PCR and sequencing using the primers B6.129_Casp11_F 5′ GTTATCTATCAGTAGGAAGTGG 3′ and B6.129_Casp11_R 5′ AAAC TAATACTTCTTATGAGAGC 3′; 129 mice have a distinguishable 5 bp deletion encompassing the exon 7 splice acceptor junction (**Kayagaki et al., 2011**). The *Hiccs* locus was genotyped by PCR using the primers D3Mit348_F 5′ CATCATGCATACTTTTTTCCTCA 3′, D3Mit348_R 5′ GCCAAATCATTCACAG CAGA 3′, D3Mit319_F 5′ TCTCCCTCACTTTTTCCTTCC 3′, and D3Mit319_R 5′ AACAGCCAGTCC AGCAAATC 3′ to distinguish polymorphisms between the B6 and 129 alleles. B6.$Nlrc4^{-/-}Casp11^{-/-}$ animals were generated by targeting *Casp11* via CRISPR-Cas9 mutagenesis in existing B6.$Nlrc4^{-/-}$ mice. CRISPR/Cas9 targeting was performed by electroporation of Cas9 protein and sgRNA into fertilized zygotes, essentially as described previously (**Chen et al., 2016**). Founder mice were geno-typed by PCR and sequencing using the primers: Casp4_F 5′ GTCTTTAGCCCTTGAGAAGGACAC 3′ and Casp4_R 5′ CACCCCTTCACTTGAGTTTCTCC 3′. Founders carrying mutations were bred one generation to B6.$Nlrc4^{-/-}$ mice to separate modified haplotypes. Homozygous lines were generated by interbreeding heterozygotes carrying matched haplotypes. Mice harboring a loxP-STOP-loxP-$Nlrc4$ transgene integrated into the *Rosa26* locus ($Rosa26^{LSL-Nlrc4}$ mice) (**Rauch et al., 2017**) were previously described. $Rosa26^{LSL-Nlrc4}$ mice were crossed to the B6.$Nlrc4^{-/-}$ line and then further crossed to $Lyz2^{Cre}$ (Jax strain 004781) transgenic lines on a B6.$Nlrc4^{-/-}$ background to generate $Nlrc4^{-/-}Rosa26^{LSL-Nlrc4}Lyz2^{Cre}$ mice that we refer to here as $iNlrc4^+Lyz2Cre^+$ mice. $Nlrc4^{-/-}Il1r1^{-/-}$ mice were generated by crossing B6.$Nlrc4^{-/-}$ mice to B6.$Il1r1^{-/-}$ mice (Jax strain 003245). B6.$Casp8^{-/-}Ripk3^{-/-}$, B6.$Casp1/11^{-/-}Ripk3^{-/-}$, and B6.$Casp1/8^{-/-}Ripk3^{-/-}$ mice were generated as previously described (**Rauch et al., 2017**).

## *Shigella* strains

Mouse infections were conducted with the *S. flexneri* serovar 2a 2457T strain, WT or *ΔospC3* (**Mou et al., 2018**). Natural streptomycin-resistant strains of WT and *ΔospC3* were generated by plating cultured bacteria on tryptic soy broth (TSB) plates containing 0.01% Congo red (CR) and increasing concentrations of streptomycin sulfate. Streptomycin-resistant strains were confirmed to grow indis-tinguishably from parental strains in TSB broth lacking antibiotics, indicating an absence of strepto-mycin dependence.

## In vivo *Shigella* infections and treatments

Streptomycin-resistant *S. flexneri* was grown at 37°C on tryptic soy agar plates containing 0.01% CR, supplemented with 100 μg/mL of streptomycin sulfate. For infections, a single CR-positive colony was inoculated into 5 mL TSB and grown shaking overnight at 37°C. Saturated cultures were back-diluted 1:100 in 5 mL fresh TSB shaking for 2–3 hr at 37°C. The approximate infectious dose was determined by spectrophotometry ($OD_{600}$ of $1=10^8$ CFU/mL). Bacteria were pelleted at 5000×*g*, washed twice in PBS, and suspended in PBS for infection by oral gavage. Actual infectious dose was determined by serially diluting a fraction of the initial inoculum and plating on TSB plates containing 0.01% CR and 100 μg/mL streptomycin. Mouse infections were performed in 6- to 22-week-old mice. Initially, mice deprived of food and water for 4–6 hr were orally gavaged with 100 μL of 250 mg/mL streptomycin sulfate dissolved in water (25 mg/mouse) and placed in a cage with fresh bedding. One day later, mice again deprived of food and water for 4–6 hr were orally gavaged with 100 μL of log-phase, streptomycin-resistant *S. flexneri* suspended in PBS at a dose of $10^8$ CFU/mL ($10^7$ CFU/mouse). Mouse weights and fecal pellets were recorded or collected daily from 1 day prior to infection to the day of euthanasia and harvest to assess the severity of disease and biomarkers of inflammation. Fecal colo-nization (CFU/g of feces) and successful challenge were determined by homogenizing feces collected 1 day post-infection and plating (see below). In rare cases when mouse feces were not colonized with *Shigella*, mice were omitted from analysis. For each mouse infection experiment, at least three mice were included in each experimental group. All mouse infection experiments were repeated at least one time (with the exception of *Figure 1*, *Figure 1—figure supplement 1*, and *Figure 1—figure supplement 2*). Blinding and randomization were applied when co-housing mice and ARRIVE guide-lines were applied when applicable. Each mouse had a unique numbered ear-tag identifier that was only associated with a treatment group or genotype following data collection. For in vivo antibody

treatments, 200 µg of anti-TNFα antibody (Bio X Cell, clone TN3-19.12) and polyclonal Armenian hamster IgG isotype control antibody (Bio X Cell) were administered by intraperitoneal injection daily starting 1 day prior to infection.

## Fecal CFUs, fecal MPO ELISAs, wet/dry ratio, fecal blood

Fecal pellets were collected in 2 mL tubes, suspended in 1 mL of 2% FBS in PBS containing protease inhibitors, and homogenized using a polytron homogenizer at 18,000 rpm. For CFU enumeration, serial dilutions were made in PBS and plated on TSB plates containing 0.01% CR and 100 mg/mL streptomycin sulfate. For MPO ELISAs, fecal homogenates were spun at 2000×$g$ and supernatants were plated in duplicate on absorbent immunoassay 96-well plates. Recombinant mouse MPO standard, MPO capture antibody, and MPO sandwich antibody were purchased from R&D Systems. Wet/dry ratios were determined by weighing fecal pellets before and after they had been dried in a fume hood. The presence or absence of fecal blood in fresh pellets was determined by macroscopic observation or by applying wet fecal samples to detection tabs from a Hemoccult blood testing kit (Beckman Coulter).

## Intestinal CFU determination

To enumerate intracellular *Shigella* CFU from the IEC fraction of mouse ceca and colons, organs were removed from mice upon sacrifice, cut longitudinally and removed of luminal contents by washing in PBS. Tissues were placed in 14 mL culture tubes, incubated in RPMI with 5% FBS, 2 mM L-glutamine, 25 mM HEPES, and 400 µg/mL of gentamicin for 1–2 hr, and vortexed briefly. Tissues were then washed five times in PBS, cut into 1 cm pieces, placed in 15 mL of stripping solution (HBSS, 10 mM HEPES, 1 mM DTT, 2.6 mM EDTA), and incubated at 37°C for 25 min with gentle agitation. Supernatants were passed through a 100 µm filter and the remaining pieces of tissue were shaken vigorously in a 50 mL conical with 10 mL of PBS and passed again through the 100 µm filter. This enriched epithelial cell fraction was incubated in 50 µg/mL gentamicin for 30–40 min on ice, spun at 300×$g$ at 4°C for 8 min, and washed twice by aspirating the supernatant, resuspending in PBS, and spinning at 300×$g$ at 4°C for 5 min. After the first wash, a fraction of cells were set aside to determine the cell count. After the second wash, the pellet was resuspended and lysed in 1 mL of 1% Triton X-100. Serial dilutions were made from this solution and plated on TSB agar plates containing 0.01% CR and 100 µg/mL streptomycin and CR+ positive colonies were counted following overnight incubation at 37°C.

## Tissue ELISAs

After isolating the IEC fraction (above), the remaining tissue was transferred to a 14 mL culture tube containing 1 mL of PBS containing 2% FBS and protease inhibitors. Organs were homogenized using a polytron homogenizer at 20,000 rpm, centrifuged at 2000×$g$, and supernatants were plated on absorbent immunoassay 96-well plates. Recombinant mouse CXCL1 and IL-1β standards, capture antibodies, and sandwich antibodies were purchased from R&D. TNFα levels were detected using a high sensitivity ELISA from Thermo Fisher (order no: BMS607HS).

## Immunoblot and antibodies

Lysates were prepared from *Casp11*$^{+/-}$ and *Casp11*$^{-/-}$ mouse bone marrow-derived macrophages and clarified by spinning at 16,100×$g$ for 10 min at 4°C. Clarified lysates were denatured in SDS loading buffer. Samples were separated on NuPAGE Bis-Tris 4–12% gradient gels (Thermo Fisher) following the manufacturer's protocol. Proteins were transferred onto Immobilon-FL PVDF membranes at 375 mA for 90 min and blocked with Odyssey blocking buffer (Li-Cor). Proteins were detected on a Li-Cor Odyssey Blot Imager using an anti-Caspase-11 primary antibody (cone 17D9) and Alex Fluor-680 conjugated secondary antibody (Invitrogen).

## Statistical analysis

Statistical significance was determined using Prism (GraphPad) software for unpaired, two-tailed Mann-Whitney test when comparing two groups, one-way or two-way ANOVA tests with Tukey's multiple comparisons test when comparing multiple groups, and Fisher's exact test when comparing categorical data (for fecal blood scores). For some ANOVA calculations, non-normal data was first log-transformed to achieve normality (see figure legends). For Fisher's exact tests,

data were stratified into two groups by presence (score = 1 or 2) or absence (score = 0) of blood. Each Fisher's exact test was performed independently between the experimental groups indicated in the figures.

## Materials availability statement

All materials used, including *Shigella* strains and mouse lines, are available on request. Please contact corresponding author Russell E Vance.

## Acknowledgements

We thank P Mitchell, I Rauch, S Fattinger, and K Eislmayr for discussion and advice. We are grateful to members of the Vance and Barton Labs for discussions. Funding: REV is an HHMI Investigator, a recipient of HHMI EPI support, and supported by NIH grants AI075039, AI155634 and AI063302. JLR is an Irving H Wiesenfeld CEND Fellow; EAT is supported by NSF GRFP DGE 1752814; CFL is a Brit d'Arbeloff MGH Research Scholar and supported by NIH AI064285 and NIH AI128743.

## Additional information

### Competing interests

Russell E Vance: Reviewing editor, *eLife*. The other authors declare that no competing interests exist.

### Funding

| Funder | Grant reference number | Author |
| --- | --- | --- |
| Howard Hughes Medical Institute | | Russell E Vance |
| National Institutes of Health | AI075039 | Russell E Vance |
| National Institutes of Health | AI063302 | Russell E Vance |
| National Institutes of Health | AI155634 | Russell E Vance |

The funders had no role in study design, data collection and interpretation, or the decision to submit the work for publication.

### Author contributions

Justin L Roncaioli, Conceptualization, Data curation, Formal analysis, Validation, Investigation, Visualization, Methodology, Writing - original draft, Writing - review and editing; Janet Peace Babirye, Roberto A Chavez, Fitty L Liu, Elizabeth A Turcotte, Investigation; Angus Y Lee, Resources; Cammie F Lesser, Resources, Supervision, Funding acquisition, Methodology, Writing - review and editing; Russell E Vance, Conceptualization, Resources, Supervision, Funding acquisition, Writing - original draft, Writing - review and editing

### Author ORCIDs

Russell E Vance (ID) http://orcid.org/0000-0002-6686-3912

### Ethics

This study was performed in strict accordance with the recommendations in the Guide for the Care and Use of Laboratory Animals of the National Institutes of Health. All of the animals were handled according to approved institutional animal care and use committee (IACUC) protocols (AUP-2014-09-6665-1) of the University of California Berkeley.

### Decision letter and Author response

Decision letter https://doi.org/10.7554/eLife.83639.sa1
Author response https://doi.org/10.7554/eLife.83639.sa2

## Additional files

### Supplementary files
• MDAR checklist

### Data availability
All data generated or analysed during this study are included in the manuscript or have been deposited with Dryad at https://doi.org/10.6078/D1S13W.

The following dataset was generated:

| Author(s) | Year | Dataset title | Dataset URL | Database and Identifier |
|---|---|---|---|---|
| Vance RE, Roncaioli J | 2023 | Data from: A hierarchy of cell death pathways confers layered resistance to shigellosis in mice | https://dx.doi.org/10.6078/D1S13W | Dryad Digital Repository, 10.6078/D1S13W |

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
