## [Editor Report]

This paper provides important new information on the role of cellular death pathways in mediating resistance and susceptibility of mice to experimental shigellosis. The results rely on experimental observations on the outcome of Shigella in mice gene deficiencies and are convincing. The results will be of interest to immunologists, cell biologists and infectious disease researchers.

---

## [Decision Letter]

**Decision letter after peer review:**

Thank you for submitting your article "A hierarchy of cell death pathways confers layered resistance to shigellosis in mice" for consideration by *eLife*. Your article has been reviewed by 3 peer reviewers, and the evaluation has been overseen by a Reviewing Editor and Arturo Casadevall as the Senior Editor. The following individual involved in review of your submission has agreed to reveal their identity: Edward A Miao (Reviewer #3).

Essential revisions:

1) Please address concerns about the rigor of the statistical analysis

2) Please address the multiple requests for clarification about experimental details and text from all three reviewers.

*Reviewer #2 (Recommendations for the authors):*

1. Is epithelial cell death further defective in Nlrc4-/-Casp11-/- and Casp1/11/8-/- Ripk3-/- mice compared to Nlrc4-/- mice? An experimental quantification of epithelial cell death in vivo would strengthen the paper, such as immunohistochemistry or immunofluorescence microscopy for cleaved caspase-3. However, this reviewer recognizes that this is potentially challenging to address in vivo, but such assays have been performed in published studies (ie PMID 31606566). Alternatively, the authors could show effects on cell death and extrusion in cultured epithelial cells, similar to their previous studies (Mitchell and Roncaioli et al. 2020,, Rauch et al., 2017).

2. The claim made in the discussion that the mechanism of TNF⍺-mediated anti-Shigella defense is "not likely driven by TNF⍺-dependent activation of NF-κB" may be overstated because caspase-8 can drive NF-κB activation and cytokine production in various settings, which the authors. Additionally, the increased bacterial burdens in caspase-8-deficient mice and mice treated with anti-TNF⍺ confound the interpretation of increased cytokine production. Experimentally, this could be addressed by measuring cell-intrinsic cytokine production by flow cytometry at a timepoint or tissue where the bacterial burdens are equal between groups. Alternatively, the authors could modify the text of the discussion by acknowledging the alternative interpretation to include the possibility that NF-κB activation downstream of TNF⍺ signaling may contribute to bacterial control.

The recommendations below would strengthen the paper but are not critical to support the conclusions of the study. Whether to address these points experimentally or by appropriate modification of the text should be left to the discretion of the authors.

1. The claim in Figure 2 that the caspase-11 inflammasome "prevents disease" in Nlrc4-deficient mice, as phrased in the figure legend seems like it may be an overstatement, or at least not nuanced enough given the data. In particular, there is no difference in weight, and just 2/9 mice had blood in the gut, making it not possible to make a robust statistical conclusion from these data. A possible explanation is contained subsequently within the paper itself, in that OspC3 efficiently blocks Casp-11 activation, likely masking anti-bacterial effects of Casp-11. Based on this, the description of Figure 2 in the text may benefit from modification in light of this possible interpretation.

2. What happens at later timepoints to Casp1/11/8-/- Ripk3-/- mice? Do they eventually recover their weight and diarrhea like previously shown in 129.Nlrc4-/- mice (Mitchell and Roncaioli et al. 2020), or do they succumb to infection? If the authors already have this data, it would further define the nature of the increased susceptibility of these mice. However, this experiment is not essential to the claims of the paper.

3. Statistical recommendations:

a. Figure 5: Please provide statistics comparing WT and iNlrc4Lyz2Cre groups in all panels.

b. For statistical analyses involving more than two groups, please use a test that corrects for multiple comparisons (particularly in Figures 4 and 8).

c. The authors may consider using a statistical test in occult blood analyses, such as a Fisher's exact test, which can analyze categorical data.

*Reviewer #3 (Recommendations for the authors):*

Introduction states that T3SS mediates actin-based motility, but I think this is primarily IcsA dependent, which I think is not a T3SS effector.

In the figure 1 legend the authors should state that this is experiment is representative of 1 experiment. We, the readers, should not infer this by omission.

Please add the "129S1/SvImJ" full strain name to the methods section.

The authors should state that CXCL1 is measured as markers of disease rather than as specific mediators to be investigated in this paper. Figure 1d shows an ELISA for IL-1β from tissue homogenate. This signal could arise from either pro-IL-1β or mature IL-1β. If the ELISA has the ability to detect both forms, then this may be a marker for NF-κB responses rather than caspase-1 activation. The authors make no claims either way. They should simply comment on this in the text, as this is not discussed currently.

Figure 1 paragraph 2: "mixed homozygous 129/129 or heterozygous B6/129 at all loci". This is not correct, please revise.

The way this data is discussed in the text, and presented in the figure legends (Figure 1 and Figure 1--supplementary figure 2) is incorrect. To my understanding, the data in these two figures are the same data but groups split different based on Casp11 vs Hiccs genotypes. The legend needs to clearly state that it is the same data. In fact, all the details of the supplementary label could be removed, simply referring to figure 1. This would make it even more clear that it is the same experimental data. This can also be facilitated by more clear writing and transition sentences in the main text.

Figure 1 should be relabeled so it is more clear that the groups "129/129 at Casp11" and "B6/129 at Casp11" are also deficient for Nlrc4. Perhaps simply writing Nlrc4-/- across the bottom with a line over all the groups? Also check panel 1G label for consistency.

The interpretation for Figure 1c-1d-1e is not correct. The authors describe the genotypes N2 cross Casp11(129/129) mice as having "modest increases" in inflammatory cytokines and cecum shrinkage, however, the CXCL1 p value is 0.09, which could be described as trending but not statistically significant, the IL-1b levels are not different by eye, and the cecum length perhaps is trending.

Figure 2E is statistically significant in the difference, but Figure 2F is only trending, so this needs to be stated correctly in the text.

Figure 3B and 3D are paired in the sentence when discussed, but 3B is significant and 3D is trending.

Figure 3F and 3G are again paired in the sentence as being "reduced levels", but 3F is significant and 3G is not even trending to different.

Figure 7A-E are described as "modest increases" however most of these are not statistically significant. There might be real differences there if the power of the experiment were doubled or tripled, however, this is probably not worth the effort. In comparison, Figure 6A has trending differences where IL-1R appears to have a trending detrimental effect on weight, but this is interpreted as not significantly different. This also might be significant with sufficient power, but again, not worth the effort to pursue that.

The authors are much more clear and precise in their interpretations of Figure 5F and 5G, this style should be applied to earlier figures. I may have missed other imprecise interpretations, please read each interpretation carefully to ensure they are correct.

Please add a statistical analysis section to the methods.

Regarding statistical analysis, in Figure 1A, the authors make an interpretation that the 129 parentals are the same as the N2 Casp11(129/129) and that the B6 parentals are the same as the N2 Casp11 (B6/129). However, the statistical analysis cannot use Mann-Whitney because 4 groups are being compared. If only day 3 was to be analyzed, then the authors need to use a one-way ANOVA. If time is added to this analysis, this is another variable added to the experiment, and requires a two-way ANOVA.

Figure 3, 4, 6, 7N legends state that a Mann-Whitney test was used for all data analyses, but the statistics show comparisons across more than two groups. A one-way ANOVA (or Kruskal-Wallis) is the correct analysis in this case.

There are no statistical analysis on the F1 mouse experiments in supplement figure 1, but conclusions are drawn from these experiments in the text so statistical analysis should be performed.

Many figures use littermates. The authors should state in the results text what the parent genotypes were. They also need to state the full genotype of each gene in the panel in the graphical figure. Heterozygous sufficiency cannot be assumed, and heterozygous states must be stated on each mouse. If all heterozygous mice were discarded and mice arise from het x het breeding, this should be stated for clarity so that a reader does not have this concern. For example in Figure 2 are the WT mice actually Nlrc4+/- and Casp11+/-, and are the Nlrc4-/- mice actually Nlrc4-/- Casp11+/- mice? Another example, in Figure 8, I can only imagine that all these mice are heterozygous at other loci that are not stated, ie the Casp8-/- Ripk3-/- mice are probably also Casp1/11+/-. The full genotype needs to be stated. Also, the legend makes it sound like the WT mice are also littermates. Are these heterozygous mice, or are they a non-co-housed WT control comparison group? This should be stated.

In the "loss of multiple cell death…" section for Figure 8, at the beginning of this section the authors have already introduced the concept that OspC3 is an incomplete inhibitor, and they have already introduced OspC1 and OspD3. Thus, my expectation in starting to read this section was that these might be incomplete inhibitors as well, but this was not mentioned. Also the discussion would benefit from direct naming and discussing these effectors. I would expect additional "genetics squared" phenotypes in future work that examined these mutants. In other words, stronger phenotypes may be observed for caspase-8 when examining an ospC1 mutant. Revision of the text would be helpful for the reader.

Fecal blood graphs use dots for each mouse, however, most graphs these dots coalesce and thus are not visible. The authors compensate by labelling the numbers of mice that have blood or not. Perhaps a different type of graph would more accurately convey the data. Perhaps diamonds would make it visible. Perhaps a y axis of percentage of mice, and then each genotype is a "stacked bar" with fecal blood scores. Or maybe a table inset in the figure?

Model in Figure Supplement 2, I think it would not take too much effort to add RIPK3, OspC1, and OspD3 to the figure, all are relevant to the manuscript. Also, if you rearrange to place the CASP11 module on the left, then you can remove the CASP8 from underneath NLRC4, instead drawing a line from NLRC4 to CASP8 that is under the TNF receptor (thus avoiding having CASP8 on the figure twice).

I suggest revising the text around the use of Casp8-/- Ripk3-/- mice. Currently, it seems that these mice only investigate caspase-8, and it is assumed that RIPK3 plays no role. However, the role for RIPK3 may only be apparent when Shigella inhibits caspase-8. Which it does with OspC1, thus explaining the existence of OspD3. Thus, most likely, both play a role, but the RIPK3 role is not apparent in a single knockout because caspase-8 is sufficient.

---

## [Author Response]

Essential revisions:1) Please address concerns about the rigor of the statistical analysis2) Please address the multiple requests for clarification about experimental details and text from all three reviewers.

We thank all the reviewers for their constructive and detailed comments. We have made numerous changes addressing the statistics and experimental details in response to the critiques of all three reviewers. These changes are detailed below and we believe they have substantially improved the manuscript.

Reviewer #2 (Recommendations for the authors):1. Is epithelial cell death further defective in Nlrc4-/-Casp11-/- and Casp1/11/8-/- Ripk3-/- mice compared to Nlrc4-/- mice? An experimental quantification of epithelial cell death in vivo would strengthen the paper, such as immunohistochemistry or immunofluorescence microscopy for cleaved caspase-3. However, this reviewer recognizes that this is potentially challenging to address in vivo, but such assays have been performed in published studies (ie PMID 31606566). Alternatively, the authors could show effects on cell death and extrusion in cultured epithelial cells, similar to their previous studies (Mitchell and Roncaioli et al. 2020,, Rauch et al., 2017).

We agree with the reviewer that an experimental quantification of cell death would strengthen the conclusions of the paper. We note that it is experimentally challenging to directly identify/quantify cell death in the intestinal epithelium in the context of *Shigella* infection. The ex vivo organoid epithelial cell system used in our previous studies was effective in part because there was a distinct and obvious difference in cell death between NLRC4-competent and NLRC4-deficient organoid monolayers, likely because NLRC4 strongly responds to a synchronized *Shigella* monolayer infection. However, the effects of CASP11 and TNF are more subtle than the effects of NLRC4 and may depend on cytokine responses that originate from hematopoietic cells (not present in organoids).

In Rauch et al. (2017), cell death was readily detectable because the administration of FlaTox allows for a synchronized, rapid, and potent cell death response. However, in the context of *Shigella* infection, although cell death appears important in limiting bacterial replication and dissemination in the epithelium, cell death events are not synchronized, occur less frequently, and are thus more difficult to visually capture. Furthermore, it is difficult to determine the pathway or sensors that initiate a given cell death pathway, as immunofluorescence for apoptotic, pyroptotic, and necroptotic markers and initiators is difficult to perform in vivo. Nonetheless, we hope to directly quantify cell death in vivo in our future work.

2. The claim made in the discussion that the mechanism of TNF⍺-mediated anti-Shigella defense is "not likely driven by TNF⍺-dependent activation of NF-κB" may be overstated because caspase-8 can drive NF-κB activation and cytokine production in various settings, which the authors. Additionally, the increased bacterial burdens in caspase-8-deficient mice and mice treated with anti-TNF⍺ confound the interpretation of increased cytokine production. Experimentally, this could be addressed by measuring cell-intrinsic cytokine production by flow cytometry at a timepoint or tissue where the bacterial burdens are equal between groups. Alternatively, the authors could modify the text of the discussion by acknowledging the alternative interpretation to include the possibility that NF-κB activation downstream of TNF⍺ signaling may contribute to bacterial control.

We agree with the reviewers that our claims regarding the role of TNF⍺ in protection are overstated in certain sections of the text. We have modified our Discussion section to include the possibility that this protection might occur through NFkB signaling and have further explained our thought process for why we hypothesize that protection is via apoptosis. We also now acknowledge the caveat that increased bacterial burdens in mice treated with anti-TNF⍺ confound the interpretation of increased cytokine production.

The recommendations below would strengthen the paper but are not critical to support the conclusions of the study. Whether to address these points experimentally or by appropriate modification of the text should be left to the discretion of the authors.1. The claim in Figure 2 that the caspase-11 inflammasome "prevents disease" in Nlrc4-deficient mice, as phrased in the figure legend seems like it may be an overstatement, or at least not nuanced enough given the data. In particular, there is no difference in weight, and just 2/9 mice had blood in the gut, making it not possible to make a robust statistical conclusion from these data. A possible explanation is contained subsequently within the paper itself, in that OspC3 efficiently blocks Casp-11 activation, likely masking anti-bacterial effects of Casp-11. Based on this, the description of Figure 2 in the text may benefit from modification in light of this possible interpretation.

We have modified the title of this section and figure to reflect the modest effect that CASP11 has on protection in the presence of OspC3. We note that the significant increases in bacterial burden in IECs (Figure 2B) and CXCL1 indicate that CASP11 does still play a minor role in protection, even if this phenotype is not fully penetrant.

2. What happens at later timepoints to Casp1/11/8-/- Ripk3-/- mice? Do they eventually recover their weight and diarrhea like previously shown in 129.Nlrc4-/- mice (Mitchell and Roncaioli et al. 2020), or do they succumb to infection? If the authors already have this data, it would further define the nature of the increased susceptibility of these mice. However, this experiment is not essential to the claims of the paper.

We have occasionally observed death in both Casp1/11/8^–/–^ Ripk3^–/–^ and 129.Nlrc4^–/–^ mice between 2 and 6 days following infection, however, we have not directly quantified these results as the purpose of these experiments were not to address survival rate in response to infection. We find that survival rate within these genotypes tends to vary between infections, suggesting that the microbiome (or other factors) might affect susceptibility to lethal infection. We intend to collect data for survival curves in future work.

3. Statistical recommendations:a. Figure 5: Please provide statistics comparing WT and iNlrc4Lyz2Cre groups in all panels.

We have used a one way ANOVA with Tukey’s multiple comparison test to analyze these data for significance.

b. For statistical analyses involving more than two groups, please use a test that corrects for multiple comparisons (particularly in Figures 4 and 8).

We have updated the manuscript to include one and two way ANOVA tests with Tukey’s multiple comparisons when analyzing more than one group in specific experiments.

c. The authors may consider using a statistical test in occult blood analyses, such as a Fisher's exact test, which can analyze categorical data.

We used a Fisher’s exact test to analyze occult blood scores (with the exception of Figure 1 and Figure 1 —figure supplement 2, which uses a one-way ANOVA to analyze significance between our added blood scores). In our Fisher’s exact analyses, data was stratified into two groups by presence (score = 1 or 2) or absence (score = 0) of blood.

Reviewer #3 (Recommendations for the authors):Introduction states that T3SS mediates actin-based motility, but I think this is primarily IcsA dependent, which I think is not a T3SS effector.

We have modified the introduction with this correction.

In the figure 1 legend the authors should state that this is experiment is representative of 1 experiment. We, the readers, should not infer this by omission.

We have modified the legend with this correction.

Please add the "129S1/SvImJ" full strain name to the methods section.

We have modified the methods section with this correction.

The authors should state that CXCL1 is measured as markers of disease rather than as specific mediators to be investigated in this paper. Figure 1d shows an ELISA for IL-1β from tissue homogenate. This signal could arise from either pro-IL-1β or mature IL-1β. If the ELISA has the ability to detect both forms, then this may be a marker for NF-κB responses rather than caspase-1 activation. The authors make no claims either way. They should simply comment on this in the text, as this is not discussed currently.

We have changed the text in the Results section to reflect that we use both IL-1b and CXCL1 as biomarkers of disease and not as specific mediators to be investigated. We also state that the ELISA does not distinguish between unprocessed and processed IL1b.

Figure 1 paragraph 2: "mixed homozygous 129/129 or heterozygous B6/129 at all loci". This is not correct, please revise.

We have corrected this in the text.

The way this data is discussed in the text, and presented in the figure legends (Figure 1 and Figure 1--supplementary figure 2) is incorrect. To my understanding, the data in these two figures are the same data but groups split different based on Casp11 vs Hiccs genotypes. The legend needs to clearly state that it is the same data. In fact, all the details of the supplementary label could be removed, simply referring to figure 1. This would make it even more clear that it is the same experimental data. This can also be facilitated by more clear writing and transition sentences in the main text.

Yes, the data in Figure 1 and Figure 1 — figure supplement 2 are the same. To clarify this, we have used more a more precise description of the experiments and data presentation in both the text and figure legends.

Figure 1 should be relabeled so it is more clear that the groups "129/129 at Casp11" and "B6/129 at Casp11" are also deficient for Nlrc4. Perhaps simply writing Nlrc4-/- across the bottom with a line over all the groups? Also check panel 1G label for consistency.

We have updated the figure legends in both Figure 1 and Figure 1 — figure supplement 2 to indicate that mice in these groups are also deficient for NLRC4. We have updated the label in panel 1G.

The interpretation for Figure 1c-1d-1e is not correct. The authors describe the genotypes N2 cross Casp11(129/129) mice as having "modest increases" in inflammatory cytokines and cecum shrinkage, however, the CXCL1 p value is 0.09, which could be described as trending but not statistically significant, the IL-1b levels are not different by eye, and the cecum length perhaps is trending.

We have changed our interpretations to reflect the reviewers comments.

Figure 2E is statistically significant in the difference, but Figure 2F is only trending, so this needs to be stated correctly in the text.

We have changed our interpretations to reflect the reviewers comments.

Figure 3B and 3D are paired in the sentence when discussed, but 3B is significant and 3D is trending.

We have changed our interpretations to reflect the reviewers comments and we note that after updating our statistical tests, data in Figure 3D are significant.

Figure 3F and 3G are again paired in the sentence as being "reduced levels", but 3F is significant and 3G is not even trending to different.

We have changed our interpretations to reflect the reviewers comments.

Figure 7A-E are described as "modest increases" however most of these are not statistically significant. There might be real differences there if the power of the experiment were doubled or tripled, however, this is probably not worth the effort. In comparison, Figure 6A has trending differences where IL-1R appears to have a trending detrimental effect on weight, but this is interpreted as not significantly different. This also might be significant with sufficient power, but again, not worth the effort to pursue that.The authors are much more clear and precise in their interpretations of Figure 5F and 5G, this style should be applied to earlier figures. I may have missed other imprecise interpretations, please read each interpretation carefully to ensure they are correct.

We have modified our interpretations and language in the Results section for each figure so that they are both more accurate and precise.

Please add a statistical analysis section to the methods.

We have added a statistical analysis section to the methods.

Regarding statistical analysis, in Figure 1A, the authors make an interpretation that the 129 parentals are the same as the N2 Casp11(129/129) and that the B6 parentals are the same as the N2 Casp11 (B6/129). However, the statistical analysis cannot use Mann-Whitney because 4 groups are being compared. If only day 3 was to be analyzed, then the authors need to use a one-way ANOVA. If time is added to this analysis, this is another variable added to the experiment, and requires a two-way ANOVA.

We have re-analyzed the data (at day three) using a one-way ANOVA and Tukey’s multiple comparisons test.

Figure 3, 4, 6, 7N legends state that a Mann-Whitney test was used for all data analyses, but the statistics show comparisons across more than two groups. A one-way ANOVA (or Kruskal-Wallis) is the correct analysis in this case.

We have re-analyzed the data for these figures using one-way and two-way ANOVAs and Tukey’s multiple comparisons test. We used log transformation for data without a normal distribution (as indicated in the figure legends).

There are no statistical analysis on the F1 mouse experiments in supplement figure 1, but conclusions are drawn from these experiments in the text so statistical analysis should be performed.

We have now performed statistical analysis on the data generated for this figure.

Many figures use littermates. The authors should state in the results text what the parent genotypes were. They also need to state the full genotype of each gene in the panel in the graphical figure. Heterozygous sufficiency cannot be assumed, and heterozygous states must be stated on each mouse. If all heterozygous mice were discarded and mice arise from het x het breeding, this should be stated for clarity so that a reader does not have this concern. For example in Figure 2 are the WT mice actually Nlrc4+/- and Casp11+/-, and are the Nlrc4-/- mice actually Nlrc4-/- Casp11+/- mice? Another example, in Figure 8, I can only imagine that all these mice are heterozygous at other loci that are not stated, ie the Casp8-/- Ripk3-/- mice are probably also Casp1/11+/-. The full genotype needs to be stated. Also, the legend makes it sound like the WT mice are also littermates. Are these heterozygous mice, or are they a non-co-housed WT control comparison group? This should be stated.

We have modified both the text, figures, and figure legends to more accurately represent the mouse genotypes used in each study and whether mice were littermates or cohoused cage-mates.

In the "loss of multiple cell death…" section for Figure 8, at the beginning of this section the authors have already introduced the concept that OspC3 is an incomplete inhibitor, and they have already introduced OspC1 and OspD3. Thus, my expectation in starting to read this section was that these might be incomplete inhibitors as well, but this was not mentioned. Also the discussion would benefit from direct naming and discussing these effectors. I would expect additional "genetics squared" phenotypes in future work that examined these mutants. In other words, stronger phenotypes may be observed for caspase-8 when examining an ospC1 mutant. Revision of the text would be helpful for the reader.

We have revised both the Results section and the Discussion section to account for the potential role of effectors OspC1 and OspD3. While OspC1 and OspD3 appear to inhibit apoptosis and necroptosis in humans, however, we hypothesize that these effectors are less effective in mice. Our data indicates that CASP8 is indeed active during Shigella infection, and plays a strong role in defense. This would indicate that this enzyme is not strongly inhibited by OspC1. Furthermore, we predict that an active CASP8 pathway would intrinsically inhibit necroptosis, thus rendering OspD3 “obsolete” in mice.

Fecal blood graphs use dots for each mouse, however, most graphs these dots coalesce and thus are not visible. The authors compensate by labelling the numbers of mice that have blood or not. Perhaps a different type of graph would more accurately convey the data. Perhaps diamonds would make it visible. Perhaps a y axis of percentage of mice, and then each genotype is a "stacked bar" with fecal blood scores. Or maybe a table inset in the figure?

We considered presenting fecal blood scores as percentage or proportion bar graphs, however, we were unable to make a figure that could separate experimental groups by color and still effectively visually distinguish the three possible blood score outcomes from each other. We also found that have a full bar for mouse groups that did not experience blood appeared misleading. We have taken the advice of Reviewer #3 and used diamonds instead of circles to represent each mouse. Diamonds appear more distinct and separate then circles, although we acknowledge that this new visually representation is not perfect. Labeling the number of mice in each group adds to the visually representation to present a more clear picture.

Model in Figure Supplement 2, I think it would not take too much effort to add RIPK3, OspC1, and OspD3 to the figure, all are relevant to the manuscript. Also, if you rearrange to place the CASP11 module on the left, then you can remove the CASP8 from underneath NLRC4, instead drawing a line from NLRC4 to CASP8 that is under the TNF receptor (thus avoiding having CASP8 on the figure twice).

Our data suggest that CASP8 is not strongly inhibited by OspC1, and thus, we predict that necroptosis is not central to protection in mice. At a minimum, we have no evidence that necroptosis, OspC1 or OspD3 are important in mice. Thus we have decided to omit RIPK3, OspC1, and OspD3 from our figure and retain the initial figure.

I suggest revising the text around the use of Casp8-/- Ripk3-/- mice. Currently, it seems that these mice only investigate caspase-8, and it is assumed that RIPK3 plays no role. However, the role for RIPK3 may only be apparent when Shigella inhibits caspase-8. Which it does with OspC1, thus explaining the existence of OspD3. Thus, most likely, both play a role, but the RIPK3 role is not apparent in a single knockout because caspase-8 is sufficient.

We have revised the Results and Discussion section to include more detailed comments on the potential roles of RIPK3, OspC1, and OspC3 (see above) and our predictions about the role of each during mouse infection.